# CompSRT: Quantization and Pruning for Image Super Resolution Transformers

## Abstract

Model compression has emerged as a way to reduce the cost of using image super resolution models by decreasing storage size and inference time. However, the gap between the best compressed models and the full precision model still remains large and a need for deeper understanding of compression theory on more performant models remains. Prior research on quantization of LLMs has shown that Hadamard transformations lead to weights and activations with reduced outlier, which leads to improved performance. We argue that while the Hadamard transform does reduce the effect of outliers, an empirical analysis on how the transform functions remains needed. By studying the distributions of weights and activations of SwinIR-light, we show with statistical analysis that lower errors is caused by the Hadamard transforms ability to reduce the ranges, and increase the proportion of values around $0$. Based on these findings, we introduce CompSRT, a more performant way to compress the image super resolution transformer network SwinIR-light. We perform Hadamard-based quantization, and we also perform scalar decomposition to introduce two additional trainable parameters. Our quantization performance statistically significantly surpasses the SOTA in metrics with gains as large as 1.53 dB, and visibly improves visual quality by reducing blurriness at all bitwidths. At 3-4 bits, to show our method is compatible with pruning for increased compression, we also prune $40\%$ of weights and show that we can achieve 6.67-15% reduction in bits per parameter with comparable performance to SOTA.

## 1 Introduction

Image super-resolution (SR), the task of reconstructing high-resolution (HR) images from low-resolution (LR) inputs, plays a critical role in diverse domains such as imagevideo enhancement Hitachi; Takeda et al. (2009); Su et al. (2011), medical imaging Yu et al. (2017); Greenspan et al. (2002); Yu et al. (2018); Robinson et al. (2017), and remote sensing Zhu et al. (2018); Murthy et al. (2014). Convolutional neural networks (CNNs) like EDSR Lim et al. (2017a), RDN Zhang et al. (2018), and SRResNet Ledig et al. (2017) have achieved high performance in these tasks but have high parameter counts. Transformer-based models like SwinIR-light have emerged as efficient alternatives, offering competitive performance with fewer parameters.

Despite being lighter than traditional CNNs, SwinIR-light still demands significant resources. Model compression addresses this issue through various methods like quantization and pruning. Quantization works by reducing parameter precision (e.g., from 32/16-bit to 2–4-bit), either during quantization-aware training (QAT), which jointly optimizes weights and quantization parameters, or post-training quantization (PTQ), which calibrates quantization on a frozen model. Pruning works through removing nodes that don't offer that much information, eliminating signal noise. Pruning methods can be structured, i.e. removing whole blocks/channels or unstructured, removing individual elements.

While most prior work focuses on either quantization or pruning with CNNs, recent PTQ methods like 2DQuant Liu et al. (2024a) and CondiQuant Liu et al. (2025) have adapted PTQ to SwinIR. However, the (SOTA) in PTQ, CondiQuant, does not provide deeper understanding of theory nor directly modifies distributions of weights and activations, which are known to be critical to quantization performance. As a result, it exhibits a notable gap from the full-precision (FP) SwinIR-light model.

Previous literature on quantization of LLMs (Ashkboos et al. (2024), Liu et al. (2024b), Federici et al. (2025), Chee et al. (2023), Sun et al. (2024), Tseng et al. (2024)) has found that outliers in weights

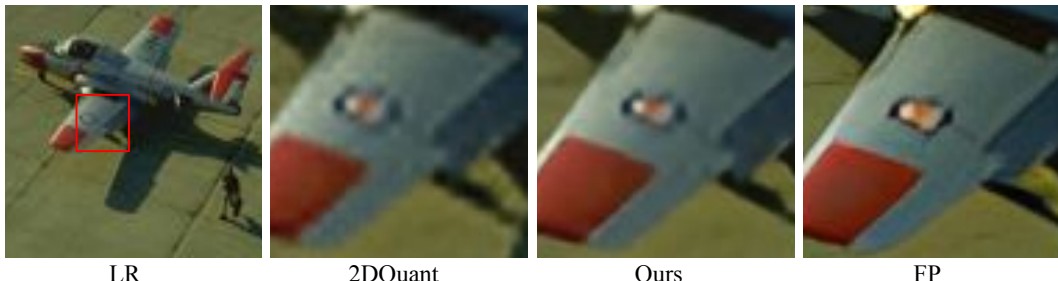

| LR | 2DQuant | Ours | FP |

Figure 1: Qualitative visual comparison for 2-bit ($\times 4$) SR on a challenging example. LR denotes low resolution image. FP denotes the output of the FP model. The comparative example is taken from 2DQuant Liu et al. (2024a)[1]. SOTA (2DQuant) suffers from excessive blurriness, while our method is significantly more clear.

and activations cause performance degradations. Sun et al. (2024) dubbed the removal of outliers as increasing the "flatness" of weights and activations and Federici et al. (2025), and Tseng et al. (2024) stated that Hadamard transformations can increase flatness of distributions. Previous arguments on how the transform functions hinge on several concepts: incoherence processing, the central limit theorem and channel mixing. Tseng et al. (2024) found theorems that Hadamard transformations concentrate the entry magnitudes of distributions in a process called incoherence processing. Liu et al. (2024b) stated that random rotations blend large and small weights together into a well-behaved distribution and empirically analyzed activation distributions via their kurtosis. Federici et al. (2025) cited the central limit theorem as to why the distributions after the Hadamard transform tend towards Gaussian and have outliers reduced. However, a detailed empirical study of both activations and weights, including statistical significance tests of how these transforms affect outliers still remains underexplored. Furthermore, combining quantization with other compression techniques in image super resolution transformers remains unexplored. Given these, our main contributions are as follows:

- Through statistical analysis on the distributions of the Swin-IR light $\times 2$ model, we show that the Hadamard transformation statistically significantly lowers the **ranges** of values which accounts for its outlier reduction and increases the **proportion of values** being concentrated specifically around **0**.

- We reparameterize the quantization scalars and zero offsets by decomposing both terms into two learnable variables, introducing two additional degrees of freedom that enable finer optimization of quantization parameters.

- We statistically significantly outperform SOTA in PSNR and SSIM across $\times 2$, $\times 3$, and $\times 4$ scale factors for all bitwidths. Specifically, we have gained **+1.53 dB** PSNR, and **+0.03** SSIM over CondiQuant Liu et al. (2025) on Manga109 at 2-bit $\times 4$. Qualitative results reveal sharper image reconstructions.

- We implement weight pruning with our quantization strategy at 3-4 bits. Using the Hadamard transform's ability to concentrate more values around 0, we prune **40%** of weights per quantized layer and have comparable performance with CondiQuant, but with **6.67%** and **15%** less bits per parameter for 3 and 4 bits respectively.

## 2 RELATED WORK

### 2.1 IMAGE SUPER RESOLUTION

EDSR (Enhanced Deep Super-Resolution) Lim et al. (2017b), is a CNN-based architecture that improves upon traditional residual networks by removing unnecessary modules and stabilizing the training procedure. SRResNet Ledig et al. (2017) employs residual learning and partial convolution based padding to generate high-quality images with fine details. SwinIR Liang et al. (2021) is a transformer-based model that has demonstrated superior performance with a reduced number of parameters compared to CNN-based approaches by leveraging shallow and deep feature extraction

---

[1]Since CondiQuant does not have open-sourced code, we compare our visual results with 2DQuant.

along with the self-attention mechanism. SwinIR is made up of Residual Swin Transformer Blocks (RSTB), which are in turn made up of Swin Transformer Layers (STL). SwinIR-light is the SwinIR model designed for lightweight SR made up of 4 RSTBs that each contain 6 STLs.

## 2.2 MODEL QUANTIZATION

Previous research has focused mainly on PTQ for CNN based architectures or vision transformers Hong & Lee (2024a); Makhov et al. (2024); Tu et al. (2023); Ding et al. (2022); Yuan et al. (2024) Li et al. (2023); Liu et al. (2023), QAT Tian et al. (2023); Hong & Lee (2024b); Hong et al. (2022a;b); Li et al. (2020); Zhong et al. (2022); Wang et al. (2021) or unique quantized architectures Qin et al. (2023). 2DQuant Liu et al. (2024a) and CondiQuant Liu et al. (2025) represent the SOTA PTQ methods for SwinIR-light. 2DQuant performs Distribution-Oriented Bound Initialization (DOBI) to search for optimal clipping ranges from input distributions, then finetunes these bounds on a calibration dataset to minimize discrepancy with the full precision model's output. CondiQuant identifies that quantization errors primarily stem from activation quantization and uses the condition number of weight matrices to measure how sensitive outputs are to small input changes, employing proximal gradient descent to minimize these condition numbers while preserving model outputs. While both methods are effective, they do not directly address large ranges in weight and activation distributions. 2DQuant initializes parameters based on distributions but doesn't modify them, while CondiQuant focuses on condition numbers rather than distribution properties. Our method goes further by reducing the ranges of the weights and activations and compacting the signal through Hadamard transforms, making distributions more quantization-friendly.

## 2.3 MODEL PRUNING

Prior work has experimented with pruning, although like quantization, the focus has been on convolutional models like EDSR. However, for transformers, Chen et al. (2023) leverage activation sparsity in window-based vision transformers to prune activations enabling speedups. Prasetyo et al. (2023) applies Sparse Regularization and Pruning methods to the Vision Transformer for image classification. Kim et al. and Jiang et al. (2023) focused on SwinIR specifically and experiment with knowledge distillation and pruning of the network. Their results show that the model compression could reduce computational costs and number of parameters without losing the performance, but their performance does not exceed ours. Lastly, Wang et al. (2025) also focus on knowledge distillation and pruning with SwinIR, letting the teacher guide channel selection during pruning. It uses a learnable, differentiable auto-pruning module and a Multiscale Wavelet Refine Module to transfer high-frequency details. While these methods are effective, none implement two methods of compression at the same time. In our work we implement both quantization and pruning for more compression at higher bitwidths. Regarding pruning criteria, we follow previous work Han et al. (2015), Lee et al. (2020), Li et al. (2018), Guo et al. (2016) in using Magnitude-based pruning, i.e. pruning the weight values with the smallest absolute magnitudes, assuming that they do not offer much information.

# 3 METHODOLOGY

## 3.1 HADAMARD TRANSFORMATION

Hadamard transformations have been used with effectiveness in quantization of LLMs to reduce the effect of outliers, which lowers errors, and make matrices tend towards the Gaussian. However, an empirical analysis on how the Hadamard functions remains underexplored, to tie it to mathematical theory. Hadamard matrices are recursively defined with entries in $\{\pm 1\}$ and implement linear, orthogonal, involutive transforms for dimensions that are powers of two, typically scaled by $1/\sqrt{n}$ (where $n$ is the last dimension) for reversibility. To implement this, we first pad each weight and activation tensor with zeros so that their dimensions become powers of two. Once padded, we perform matrix multiplication between the tensors and a Hadamard matrix of the appropriate size in full precision. This is given in $X' = (H \cdot X)/\sqrt{\dim(X)}$ where $\dim(X)$ returns the last dimension of $X$.

To empirically tie the behavior of the transform to prior theory, we statistically test whether the weight and activation distributions are more normally distributed after the transformation. We perform all of our tests on the SwinIR-light $\times 2$ model. To perform all of our statistical tests, since tensor

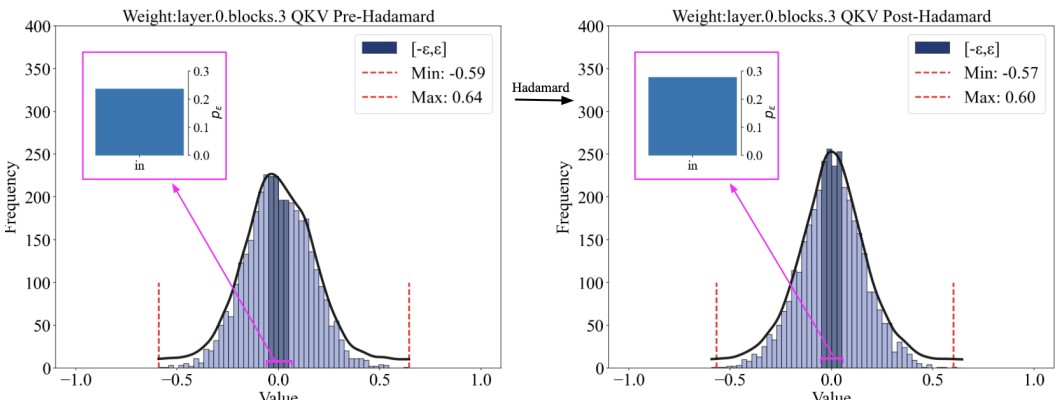

Figure 2: The left histogram shows the weight distribution prior Hadamard with a larger range and a sharper peak. The right histogram shows that the Post-Hadamard weight distribution is more Gaussian. The dark blue region indicates the values within $[-\epsilon, \epsilon]$ which now have more concentration after the transform as shown by the increase in the probability of a value being in $[-\varepsilon, \varepsilon]$, $p_\varepsilon$.

elements are not individually matchable across the Hadamard because each post-coordinate is a linear combination of many pre-coordinates, we treat each whole tensor as the experimental unit and form paired pre/post Hadamard summaries on the same tensor. For all experiments, we have 144 pairs of weight tensors and 240 pairs of activation tensors. For every paired pre/post tensor $(p_j, q_j)$, we flatten the matrices, randomly sample 1 million elements and calculate the Shapiro-Wilk $W$-statistic prior to and after the transformation. We then calculate $\Delta_{wj} = W_{qj} - W_{pj}$ (positive means more normal). Aggregating over matrices $j = 1, \ldots, N$, we use a one-sided Wilcoxon signed-rank test with $H_0 : \mathrm{median}(\Delta_w) = 0$ vs. $H_1 : \mathrm{median}(\Delta_w) > 0$, and our findings in Table 1 show that $W$ statistically significantly increases, so distributions become statistically significantly more normal after the transformation. Plotting a distribution also confirms this as shown in Figure 2.

Now, to explore empirically how the Hadamard concentrates entry magnitudes and why that is beneficial for quantization, we show that the Hadamard transformations reduce quantization errors in matrices by reducing the ranges of the values, and concentrating values around 0. This reduces the error because if values are closer together and closer to 0, quantizing them to a fixed value incurs less errors. We perform statistical tests to measure whether the Hadamard does reduce ranges and concentrate more values around 0.

To test whether the Hadamard transform statistically reduces the range of values in activations and weights, for every paired pre/post tensor $(p_j, q_j)$, we flatten the arrays and align dimensions by right-padding the pre tensor with zeros to the post length, ensuring summaries live in the same ambient space as the transform. We then compute per-tensor ranges $R_j^{\mathrm{pre}} = \max(p_j) - \min(p_j)$ and $R_j^{\mathrm{post}} = \max(q_j) - \min(q_j)$, and form paired differences $\Delta_{rj} = R_j^{\mathrm{pre}} - R_j^{\mathrm{post}}$ (positive means a reduction). Aggregating over matrices $j = 1, \ldots, N$, we assess normality of $\{\Delta_{rj}\}$ using a Shapiro–Wilk test. Finding non-normality, we use a one-sided Wilcoxon signed-rank test with ties dropped to test $H_0 : \mathrm{median}(\Delta_r) = 0$ vs. $H_1 : \mathrm{median}(\Delta_r) > 0$. For interpretability, we compute a paired effect size, Cohen's $d_z = \bar{\Delta}_r / s_{\Delta_r}$. Table 1, shows both activations and weights have statistically significant range reductions at $\alpha = 0.05$.

To test whether the Hadamard transform increases mass near zero, we evaluate the proportion of entries within $[-\varepsilon, \varepsilon]$. In our work, we set $\varepsilon = 0.05$ to be close to 0. We evaluate whether other values of $\epsilon$ will affect the statistical significance of this analysis in Section 7.1. For each pre/post pair $(p_j, q_j)$, we compute per-tensor in-band proportions $\hat{p}_j^{\mathrm{pre}} = \frac{1}{n_{pj}} \sum_{i=1}^{n_{pj}} \mathbf{1}\{|p_{j,i}| \le \varepsilon\}$ and $\hat{p}_j^{\mathrm{post}} = \frac{1}{n_{qj}} \sum_{i=1}^{n_{qj}} \mathbf{1}\{|q_{j,i}| \le \varepsilon\}$, then form the paired difference $\Delta_{pj} = \hat{p}_j^{\mathrm{post}} - \hat{p}_j^{\mathrm{pre}}$. Aggregating over tensors $j = 1, \ldots, N$ (weights and activations analyzed separately), we test $H_0 : \mathrm{median}(\Delta_p) = 0$ vs $H_1 : \mathrm{median}(\Delta_p) > 0$ using a one-sided Wilcoxon signed-rank test with ties dropped. We also compute a paired effect size (Cohen's $d_z$) on $\{\Delta_{pj}\}$.

As shown in Table 1, the $\varepsilon$-band proportion increases significantly after the Hadamard, consistent with shrinkage toward zero at $\alpha = 0.05$ with a significant effect size. All of these findings are

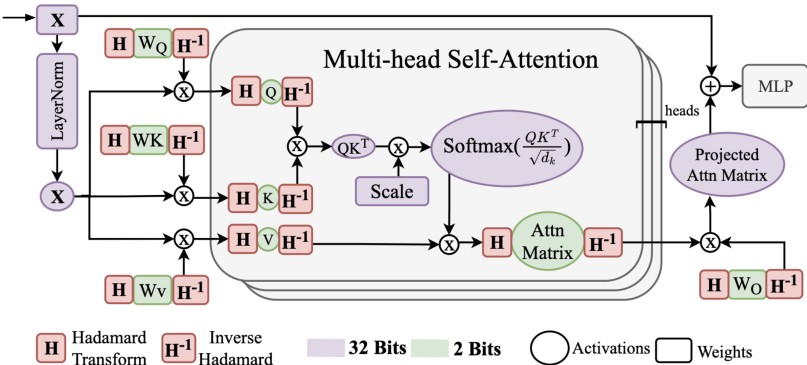

Figure 3: Architecture and Quantization scheme for Swin Transformer Layer (STL). X denotes the input. The quantized weights & activations per component are in green and the FP operations are in purple. The Hadamard and inverse Hadamard transforms are shown with red boxes.

further validated in Figure 2. The moving in of the red bars and lowering of min and max indicates range reduction and the bar graph to the left which calculates the proportion of values within the $[-\varepsilon, \varepsilon]$ band, $p_\varepsilon$, shows that $p_\varepsilon$ increased after the transformation. Overall, we find that Hadamards statistically significantly make distributions more normal, reduce the range, shrink values, move more values closer within $[-\varepsilon, \varepsilon]$.

### 3.2 SCALAR DECOMPOSITION

We quantize attention layers, linear layers and batch matrix multiplication weights and activations. The full pipeline of our method and what is quantized is shown in Figure 3 and Figure 4. To quantize the parameters to 2-4 bits, we use fake-quantization Jacob et al. (2017) i.e. quantization dequantization that simulates the loss of information through the quantization process by restricting the values to be representable by 2-4 bits but then immediately reverting them back to floats. The standard approach to fake quantization involves determining the quantization scalar, $S = \frac{u-l}{2^b-1}$, based on the upper and lower bounds of the clipping range. The equations are as follows: $v_c = \text{Clip}\,(x, l, u)$, $v_q = \text{Round}\,(\frac{2^b-1}{u-l}\,(v_c - l))$, $v_{deq} = \frac{u-l}{2^b-1}\,v_r + l$, where $x$ is the tensor to be quantized, $b$ is the bit-width, $l$ is the lower bound, and $u$ is the upper bound. The lower bound represents the zero offset, which is used to offset the quantization range, ensuring that the smallest value is 0 and so zero is exactly representable by an integer in the quantized range. In our work, we modify this value and the quantization scalar $S$ which is $\frac{u-l}{2^b-1}$. Following 2DQuant, we have a step for searching for the upper and lower bounds of the clipping range to minimize quantization error, and a step for finetuning the quantization parameters given those starting points. We additively decompose the zero offset $l$ and the quantization scalar $S = \frac{u-l}{2^b-1}$ into $S' = S + \alpha$ and $l' = l + \beta$ to be able to fine-tune each by one additional parameter. The idea behind introducing an additional predictor for the quantization scalar and zero offset serves two complementary purposes.

First, both variables are crucial for quantization, necessitating a method with high representation capabilities. By expanding the representational capacity of the model, the added predictor helps reduce the bias inherent in simpler models that rely on fewer predictors. Second, the additional parameters provide an alternative pathway through which gradients can flow more freely during backpropagation which allows for better optimization. To this end, we introduce $\alpha$, which is initialized at 0 and adjusts the quantization scalar $S$, and $\beta$ which is also initialized at 0 and we add the additional parameters to the scalar and zero point as shown above. When we search for the upper and lower bounds of the clipping range, we set these values to 0. However, during the parameter finetuning phase, we finetune these parameters starting from 0. With these adjustments, the quantization process then proceeds as follows with the clipping step omitted for brevity: $v_q = \text{Round}\,((\frac{2^b-1}{u-l} + \alpha)\,(v_c - (l + \beta)))$, $v_{deq} = (\frac{u-l}{2^b-1} + \alpha)v_q + (l + \beta)$. To see the effect of these additional parameters, we run an ablation on the 2-bit $\times 2$ model where we only learn our parameters given in Section 7.2. We see that our scalars uniquely give improvements.

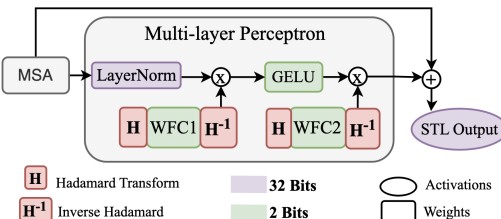

Figure 4: Architecture and quantization scheme for Swin Transformer Layer MLP. The Hadamard transformation is in full precision.

| Test | Weights | | Acts | |
|---|---|---|---|---|
| | $p$ | $d_z$ | $p$ | $d_z$ |
| $\mathrm{median}(\Delta_w) > 0$ | $3.9e^{-10}$ | 0.51 | $1.09e^{-6}$ | 0.31 |
| $\mathrm{median}(\Delta_r) > 0$ | $7.4e^{-13}$ | 0.44 | $1.6e^{-36}$ | 0.97 |
| $\mathrm{median}(\Delta_p) > 0$ | $3.0e^{-25}$ | 2.35 | $1.2e^{-4}$ | 0.61 |

Table 1: Combined statistical tests for weights and activations. For all tests, $N_{\mathrm{weights}} = 144$, $N_{\mathrm{activations}} = 240$. To measure the effect size, we report Cohen's $d_z$. All $p < 0.05$.

### 3.3 PRUNING

To achieve more compression at 3 and 4 bits, prior to quantization but after the Hadamard transformation on each weight, we perform Magnitude-based pruning on the weights. Magnitude-based pruning suggests that weights with small magnitudes do not offer much information, and as shown in Figure 2, the transform allows for more values to be closer to 0, allowing us to remove them as they will have smallest absolute magnitudes to compress the weights further. We prune 40% of weights as this is the cutoff to actually give storage gains at 3 and 4 bits and results in minimal performance degradation. To perform this pruning, we take a sample of the matrix, take the absolute value of it, and calculate the desired pruning percentile, $T_{n\%}$. Then, we set to 0 all values whose absolute values are below the threshold, corresponding to the n percent of the matrix with the smallest absolute magnitude. This follows the equation: $X_P = X \odot \mathbf{1}_{\{|X| \geq T_{n\%}\}}$, where $X_P$ is the pruned tensor, and $X$ is the input tensor. We compare this with pruning of whole channels with lowest average absolute magnitudes in Section 7.3. To actually lower bits per parameter by this pruning, we propose storing the smaller, pruned weight matrices along with a 1-bit per element mask matrix of 1s and 0s the size of the original tensor to store the indices of the pruned values in the full-sized matrix. We do not prune the 2-bit model, as this will not give any storage gains when accounting for the 1-bit mask. In Section 4.6, we show that we can gain bits per parameter by performing our method.

## 4 EXPERIMENTS

### 4.1 DATA AND EVALUATION

We use DF2K Timofte et al. (2017); Lim et al. (2017a) as the training data, which combines DIV2K Timofte et al. (2017) and Flickr2K Lim et al. (2017a). We use the Set5 Bevilacqua et al. (2012) as the validation set. We test our method on five commonly used benchmarks in the SR field: Set5 Bevilacqua et al. (2012), Set14 Zeyde et al. (2012), B100 Martin et al. (2001), Urban100 Huang et al. (2015), and Manga109 Matsui et al. (2017). The evaluation metrics we used are Peak signal-to-noise ratio (PSNR) and structural similarity index measure (SSIM) Wang et al. (2004), which are calculated on the Y channel (i.e., luminance) of the YCbCr space. For both metrics, higher indicates better performance. The implementation details of our method are given in Section 7.4.

### 4.2 RESULTS

CompSRT demonstrates superior performance to SOTA across all experimental configurations and benchmarks. Table 2 presents comprehensive comparisons of our quantization method and previous SOTA across scale factor ($\times 4$) and 2-4 bit widths. Additional results for other scale factors are included in Section 7.5. To assess whether CompSRT statistically significantly outperforms CondiQuant, for a fixed scale and bit width and for each metric $m \in \{\mathrm{PSNR}, \mathrm{SSIM}\}$, we form per-dataset differences $\Delta_i = \mathrm{score}^{(m)}_{\mathrm{CompSRT},i} - \mathrm{score}^{(m)}_{\mathrm{CondiQuant},i}$ for $i = 1, \ldots, N$ (with $N = 5$ datasets). We then test $H_0 : \mathrm{median}(\Delta) = 0$ versus $H_1 : \mathrm{median}(\Delta) > 0$ using the paired Wilcoxon signed-rank test; exact ties ($\Delta_i = 0$) are excluded. We present the one-sided Wilcoxon $p$-value and the effect size given by Cohen's $d_z = \bar{\Delta}/s_\Delta$, where $\bar{\Delta}$ is the mean of the paired differences and $s_\Delta$ is their sample standard deviation. This procedure is run independently for each (scale, bit, metric) configuration using only dataset-level PSNR and SSIM averages. As presented in Table 5, the

| Method (×4) | Bit | Set5 | | Set14 | | B100 | | Urban100 | | Manga109 | |
|---|---|---|---|---|---|---|---|---|---|---|---|
| | | PSNR | SSIM | PSNR | SSIM | PSNR | SSIM | PSNR | SSIM | PSNR | SSIM |
| SwinIR-light | 32 | 32.45 | 0.8976 | 28.77 | 0.7858 | 27.69 | 0.7406 | 26.48 | 0.7980 | 30.92 | 0.9150 |
| Bicubic | 32 | 27.56 | 0.7896 | 25.51 | 0.6820 | 25.54 | 0.6466 | 22.68 | 0.6352 | 24.19 | 0.7670 |
| PTQ4ViT | 4 | 31.49 | 0.8831 | 28.04 | 0.7680 | 27.20 | 0.7240 | 25.53 | 0.7660 | 29.52 | 0.8940 |
| NoisyQuant | 4 | 31.09 | 0.8751 | 27.75 | 0.7591 | 26.91 | 0.7151 | 25.07 | 0.7500 | 28.96 | 0.8820 |
| 2DQuant | 4 | 31.77 | 0.8867 | 28.30 | 0.7733 | 27.37 | 0.7278 | 25.71 | 0.7712 | 29.71 | 0.8972 |
| CondiQuant | 4 | 32.09 | 0.8923 | 28.50 | 0.7792 | 27.52 | 0.7345 | 25.97 | 0.7831 | 30.16 | 0.9054 |
| CompSRT (ours) | 4 | **32.41** | **0.8969** | **28.74** | **0.7849** | **27.68** | **0.7399** | **26.39** | **0.7953** | **30.81** | **0.9131** |
| PTQ4ViT | 3 | 29.77 | 0.8337 | 27.00 | 0.7248 | 26.21 | 0.6735 | 24.22 | 0.6983 | 27.94 | 0.8479 |
| NoisyQuant | 3 | 28.90 | 0.7972 | 26.50 | 0.6970 | 26.16 | 0.6628 | 23.86 | 0.6667 | 27.17 | 0.8116 |
| 2DQuant | 3 | 30.90 | 0.8704 | 27.75 | 0.7571 | 26.99 | 0.7126 | 24.85 | 0.7355 | 28.21 | 0.8683 |
| CondiQuant | 3 | 31.62 | 0.8855 | 28.20 | 0.7715 | 27.31 | 0.7269 | 25.39 | 0.7624 | 29.29 | 0.8915 |
| CompSRT (ours) | 3 | **32.31** | **0.8956** | **28.69** | **0.7839** | **27.64** | **0.7387** | **26.27** | **0.7918** | **30.60** | **0.9108** |
| PTQ4ViT | 2 | 27.23 | 0.6702 | 25.38 | 0.5914 | 25.15 | 0.5621 | 22.94 | 0.5587 | 24.66 | 0.6132 |
| NoisyQuant | 2 | 25.94 | 0.5862 | 24.33 | 0.5067 | 24.16 | 0.4718 | 22.32 | 0.4841 | 23.82 | 0.5403 |
| 2DQuant | 2 | 29.53 | 0.8372 | 26.86 | 0.7322 | 26.46 | 0.6927 | 23.84 | 0.6912 | 26.07 | 0.8163 |
| CondiQuant | 2 | 30.64 | 0.8671 | 27.59 | 0.7567 | 26.93 | 0.7136 | 24.54 | 0.7282 | 27.67 | 0.8613 |
| CompSRT (ours) | 2 | **31.44** | **0.8820** | **28.15** | **0.7696** | **27.28** | **0.7253** | **25.38** | **0.7585** | **29.20** | **0.8881** |

Table 2: Performance comparison with state-of-the-art methods for scale factor (×4) across different bit widths. All comparative results are taken from SwinIR-light Liang et al. (2021), PTQ4VIT Yuan et al. (2024), NoisyQuant Liu et al. (2023), 2DQuant Liu et al. (2024a), and CondiQuant Liu et al. (2025)as reported in their papers. Our method achieves superior performance across all configurations.

resulting p-values for all pairwise comparisons fall below the significance threshold of $\alpha = 0.05$. This allows us to reject the null hypothesis and confirms that CompSRT statistically significantly outperforms CondiQuant across all evaluated conditions. Notably, our 4-bit quantized model delivers performance remarkably close to the full-precision baseline. For (×4), the difference ranges from 0.01–0.11 dB across datasets (0.04 on Set5). These results highlight that CompSRT achieves almost full-precision quality for 4-bits while reducing the model size by a factor of 8 times. Furthermore, we have gained +1.53 dB PSNR, and +0.03 SSIM over CondiQuant Liu et al. (2025) on Manga109 at 2-bit ×4, higlighting our large gains. For generalizability, we also apply our method on Mambav2IR-light Guo et al. (2024). Results for this for all datasets are given in Section 7.7.

### 4.3 QUALITATIVE RESULTS

We show the visual results comparing the performance of our 2-bit (×4) model with the 2-bit (×4) 2DQuant model, using the original low-quality image as input, as illustrated in Figure 5. These qualitative comparisons highlight the effectiveness of our approach in enhancement. Across both natural images and manga illustrations, our method consistently outperforms 2DQuant by generating images that exhibit significantly less blurriness, more clarity and sharper lines. For example, in examples from Manga109, our method produces less grainy images, making our method applicable to animation. In images from Urban100, our model's output exhibits the lines sharply, while 2DQuant's rendering has blur. On the Set14 image with writing, our method more clearly enhances words. Lastly, As shown by the images of the man from B100 and the woman from Set5, our method can more clearly enhance close ups of human eyes, making our method more applicable to security and face identification domains.

### 4.4 PRUNING RESULTS

We show the performance of adding weight pruning to our CompSRT method on the 3 and 4-bit quantized ×4 SwinIR models on Set5. We add this step after the Hadamard transformation but prior to quantization. We experiment with varying percentages of pruning, with the results for 0% to 99.5% being show in Figure 7. The figure shows that performance in terms of PSNR and SSIM degrades after pruning any percentage for 4 bits, but at 3 bits, the performance slightly increases from 10 to 30%. This is because the small values close to 0 might have been closer to noise, and for a model with less representational capacity, setting them to 0 allows for learning a smoother more easily representable signal during finetuning. However, as the representational capacity of the 4-bit model

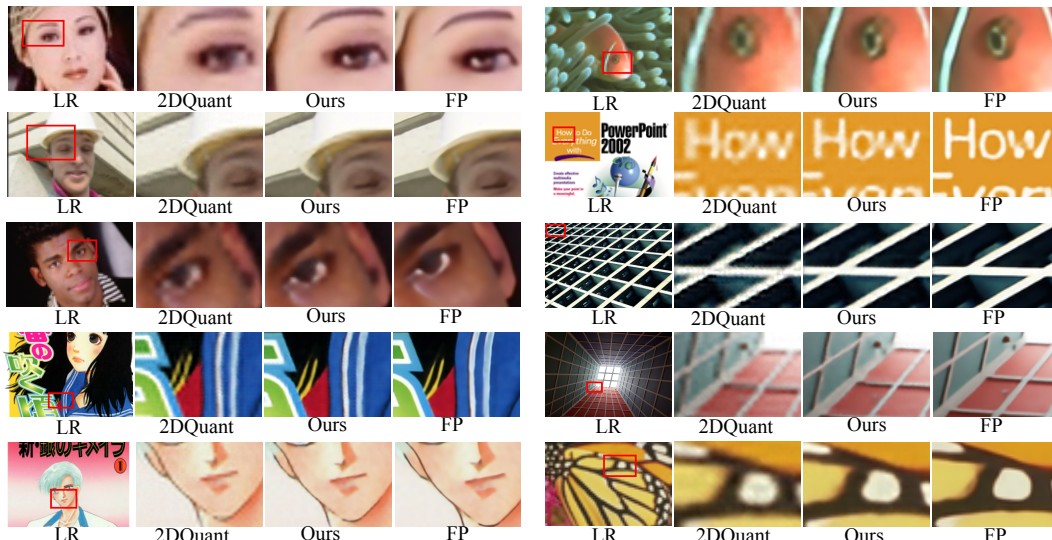

Figure 5: Qualitative visual comparison for 2-bit (×4) SR across all five benchmark datasets. LR denotes the low resolution image. Comparative examples are taken from the FP model SwinIR-light Liang et al. (2021), and 2-bit (×4) 2DQuant Liu et al. (2024a).

is exponentially larger than the 3-bit model, the 4-bit model suffers after the pruning. After 40% for both bitwidths, there is a slight drop, but our performance is comparable with SOTA CondiQuant. That is why we chose 40% pruning in our method; to achieve maximum space reduction without any cost in performance. To further explore the performance of the 40% weight pruned model, we examine the specifics of its performance across varying quantization bitwidths given in Table 3. We find that performance degraded from our quantization method by 0.39 dB for 4 bits and by 0.47 dB for 3 bits, but stayed on par with Condiquant in terms of PSNR and SSIM. This shows that 40% of the weight signal can be extraneous and can be removed without a great loss in performance. The loss at higher percentages of pruning is more pronounced, with any percentage after 40 performing worse, as show in Figure 7.

## 4.5 ABLATION STUDIES

We conduct ablation studies with the two parts of our quantization method to find the element with the most impact. The results presented in Figure 6 indicate that the most significant performance improvements stem from incorporating additional trainable parameters via scalar decomposition. The trainable parameters alone contributed to 0.64 dB increase in Set5 PSNR over the CondiQuantLiu et al. (2025) baseline. Applying Hadamard transformations to both weights and activations also yields a 0.62 dB gain over the 2DQuant baseline, proving its standalone efficacy. Furthermore, applying the Hadamard transformation to only the weights or adding trainable parameters with Hadamard transforms only on the weights did not lead to large improvements, indicating that weights benefit less from range reduction and compaction. Given these results, our analysis supports the conclusion that the primary advantage of the Hadamard transformation lies in its ability to reduce extreme values in activation distributions.

## 4.6 MODEL COMPLEXITY

We evaluate the time and space complexity of our quantization and pruning method. To evaluate the time complexity, we measured the time required to complete a single forward pass on our 4-bit quantized ×4 model and our 4-bit quantized and pruned ×4 model on an image from the Set5 benchmark dataset in seconds. We also measured the size of the model's trainable parameters in megabytes (MB) and in bits per parameter $b$, comparing with 2DQuant. Table 4 shows these results for the 4 bit ×4 model, but Table 11 in the supplementary materials have these results for the other configurations. To calculate the size for pruned models, we take into account the unpruned portion of the model, along with the storage cost of storing a 1-bit mask per matrix to store the location of 0s.

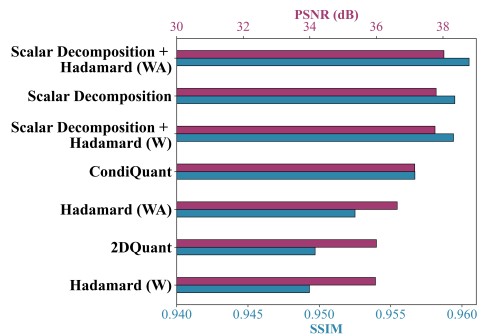

Figure 6: Ablation studies. Blue bars show SSIM and purple bars show PSNR. W = weights; WA = weights+activations. All models are ($\times 2$), 2-bit PTQ, evaluated on Set5.

Figure 7: 3 and 4-bit ($\times 4$) model PSNR and SSIM performance on Set5 vs. pruning percentage. Only quantized layers are pruned. Between 10-30% error lowers, but increases after 40%.

| Bits | PSNR | SSIM |
|------|-------|------|
| 4 | 32.02 | 0.89 |
| 3 | 31.74 | 0.89 |

Table 3: Bitwise performance for CompSRT + 40% weight-pruned ($\times 4$) models on Set5. Performance is comparable with SOTA.

| Method | Bit | $s$ | MB | $b$ |
|--------|-----|-------|-------|-----|
| 2DQuant | 4 | 0.048 | 13.99 | 4 |
| Ours | 4 | 0.092 | 14.00 | 4 |
| Ours + Prune | 4 | 0.092 | 13.95 | 3.4 |

Table 4: Time and space complexity for ($\times 4$) models. All models evaluated on Set5. $b$ denotes bits per parameter and $s$ seconds.

| Scale, Bit | PSNR | | SSIM | |
|------------|------|------|------|------|
| | $p$ | $d_z$ | $p$ | $d_z$ |
| $\times 2$, 4 | 0.031 | 1.57 | 0.031 | 1.28 |
| $\times 2$, 3 | 0.031 | 1.64 | 0.031 | 1.15 |
| $\times 2$, 2 | 0.031 | 1.83 | 0.031 | 1.11 |
| $\times 3$, 4 | 0.031 | 1.63 | 0.031 | 1.68 |
| $\times 3$, 3 | 0.031 | 1.79 | 0.031 | 1.60 |
| $\times 3$, 2 | 0.031 | 1.85 | 0.031 | 1.87 |
| $\times 4$, 4 | 0.031 | 1.89 | 0.031 | 2.33 |
| $\times 4$, 3 | 0.031 | 1.95 | 0.031 | 2.08 |
| $\times 4$, 2 | 0.031 | 1.83 | 0.031 | 2.25 |

Table 5: One-sided Wilcoxon signed-rank tests (CompSRT > CondiQuant) and paired effect sizes (Cohen's $d_z$). All $p$-values $< 0.05$.

The calculation for bits per parameter in this case at 4 bits is $(4 * 0.6 + 1) = 3.4$ vs. 4 (15% reduction in bits per parameter). The calculation for 3 bits is $(3 * 0.6 + 1) = 2.8$ vs. 3 (6.67% reduction in bits per parameter). Furthermore, Table 4 shows that our quantization method adds no storage overhead but incorporating Hadamard transformations introduces a modest computational overhead of 0.044 seconds compared to the 2DQuant baseline. This is found for all scale factors and bitwidths as shown in Table 11. However, this additional latency is expected, as this process introduces an additional dense matrix multiplication per quantized matrix. However, there is very minimal additional storage overhead because Hadamard matrices don't need to be stored and can be created when needed and we add only 2 extra parameters. When comparing pruning against 2DQuant, our approach introduces more compression but adds no additional runtime to our method, as pruning is done once per layer. This demonstrates that Hadamards can be applied to both quantization and pruning of Swin-IR light.

# 5 DISCUSSION

## 5.1 CONCLUSION

In this work, we propose a novel Hadamard guided approach to improve image SR PTQ. We challenge previous intuitions about the Hadamard transform and find that the Hadamard transform does not make distributions flatter in SwinIR-light. Instead, we hypothesized that there is another mechanism for their function in improving quantization. We find that instead it decreases the ranges and increases the concentration of values around 0 in activations and weights, which is why it lowers quantization errors. These properties improve quantization to low bit widths but also allow us to prune 40% of weights for increased compression without significant loss in performance. We also perform parameter decomposition, which leverages the bias-variance tradeoff. Our quantization method achieves statistically significant gains in quantitative metrics and visible improvements in visual quality over the SOTA quantization method, with minimal storage overhead; our pruned models have comparable performance with SOTA but with 6.67-15% less bits per parameter.

## 6  REPRODUCIBILITY STATEMENT

We have taken several steps to ensure the reproducibility of our results. The model architecture is described in Sections 2 and 3 of the main paper, and the training details are described in Section 4. All datasets used in our experiments are publicly available and evaluation methods are also described in Section 4. Finally, an anonymous link to our source code and scripts for training, evaluation, and statistical tests, together with exact run commands and an environment specification, is provided in the supplementary materials to facilitate faithful replication of our experiments.

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

## 7 SUPPLEMENTARY MATERIALS

### 7.1 $\epsilon$-SENSITIVITY OF ANALYSIS

To see whether we can still observe a statistically significant increase in the proportion of values in the $[-\epsilon, \epsilon]$ region as we vary the values of $\epsilon$, we repeated our analysis with $\epsilon \in [0.01, 1]$, testing $\mathrm{median}(\Delta_p) > 0$ for each $0.1$ increment. The results are included in this table. As we increase the value of epsilon, we see larger p-values and smaller effect sizes, which makes sense as many of these matrices are distributed around 0 with small standard deviations. However, many p-values are statistically significant, making this analysis not very sensitive to $\epsilon$.

| $\epsilon$ | Weights | | Acts | |
|---|---|---|---|---|
| | $p$ | $d_z$ | $p$ | $d_z$ |
| 0.01 | $1.1e^{-25}$ | 8.54 | $2.0e^{-8}$ | 0.72 |
| 0.1 | $5.0e^{-22}$ | 1.38 | 0.0006 | 0.57 |
| 0.2 | $2.5e^{-21}$ | 1.00 | 0.22 | 0.25 |
| 0.3 | $1.2e^{-21}$ | 0.98 | 0.15 | 0.33 |
| 0.4 | $2.8e^{-20}$ | 0.80 | 0.08 | 0.33 |
| 0.5 | $5.6e^{-13}$ | 0.43 | 0.01 | 0.29 |
| 0.6 | $2.5e^{-7}$ | 0.26 | 0.01 | 0.24 |
| 0.7 | $1.7e^{-6}$ | 0.23 | 0.004 | 0.20 |
| 0.8 | $9.9e^{-5}$ | 0.2 | 0.004 | 0.18 |
| 0.9 | 0.001 | 0.18 | .004 | 0.16 |
| 1 | 0.01 | 0.17 | 0.01 | 0.15 |

Table 6: Sensitivity of statistical test to the value of $\epsilon$. As we increase $\epsilon$, we have higher p-values and lower effect sizes, but most are statistically significant, showing that the analysis is robust to $\epsilon$.

### 7.2 SCALAR ABLATION STUDY

To isolate the effects of our additional parameters, we provide an ablation study where we do not learn the upper and lower bounds of the clipping range like 2DQuant, and instead only learn our parameters on the 2-bit $\times 2$ model, keeping everything else like the Hadamard transformations the same and evaluating on Set5. The results are given Table 7. From Figure 6 we saw that the Hadamard transforms on their own do not surpass the performance of CondiQuant. We also see that in Table 7 where we see that only learning the parameters and not learning the upper and lower bounds also does not improve performance surpassing CondiQuant, but it is better than the baseline 2DQuant, showing that our parameters are uniquely beneficial. However, they must be combined with learning the upper and lower bounds of the clipping range, which is not learning the quantization scalar or zero points to have the combined effect of surpassing CondiQuant and achieving the SOTA performance.

| Method ($\times 4$) | Bit | PSNR | SSIM |
|---|---|---|---|
| 2DQuant | 2 | 36.00 | 0.9497 |
| CondiQuant | 2 | 37.15 | 0.9567 |
| Ours- $\alpha, \beta$ tuning only | 2 | 36.50 | 0.9519 |
| Ours- tuning all | 2 | 38.03 | 0.9605 |

Table 7: Ablation study comparing effect of learned upper and lower bounds of clipping range versus only learning $\alpha$ and $\beta$ on Set5. We see that learning $\alpha$ and $\beta$ uniquely improves performance.

### 7.3 STRUCTURED CHANNEL PRUNING

We compare our unstructured magnitude based pruning with a structured channelwise pruning scheme on the weights that prunes whole channels with the lowest absolute magnitudes to achieve the desired pruning percentage. This method results in more performance degradation, which is why we did not

focus on it in our work, but offers more size compression and more hardware compatibility. We report the performance for pruning 40% of weights in quantized SwinIR-light ×4 model for all bitwidths in Table 8. Since we are not storing a dense 1-bit mask, we perform pruning for the 2-bit models as well. We also evaluate inference time in seconds, and note that structured pruning is slower to calculate than unstructured pruning in this case, as inference times increase than our unstructured pruning method. Therefore, while the structured pruning achieves more compression, we did not focus on it as the performance degradation results in performance that is below previous SOTA.

| Method (×4) | Bit | $b$ | MB | $s$ | PSNR | SSIM |
|---|---|---|---|---|---|---|
| Ours | 4 | 3.4 | 13.95 | 0.092 | 32.02 | 0.89 |
| Ours | 4 | 2.4 | 13.86 | 0.12 | 29.92 | 0.8471 |
| Ours | 3 | 2.8 | 13.90 | 0.088 | 31.74 | 0.89 |
| Ours | 3 | 1.8 | 13.82 | 0.13 | 29.90 | 0.8462 |
| Ours | 2 | 2 | 13.83 | 0.09 | 31.44 | 0.8820 |
| Ours | 2 | 1.2 | 13.77 | 0.12 | 29.84 | 0.8460 |

Table 8: Results for structured pruning of the ×4 SwinIR-light model across all quantization bitwidths on Set5. Structured pruning results in more performance degradation.

## 7.4 IMPLEMENTATION DETAILS

We build our work on top of 2DQuant's open sourced repository Liu et al. (2024a), and use SwinIR-light Liang et al. (2021) as the model backbone of our method. For the Hadamard transform and statistical tests, we use SciPy Virtanen et al. (2020). For finetuning, we use the Adam Kinga et al. (2015) optimizer with a learning rate of $1 * 10^{-2}$ and betas set to (0.9, 0.999). We clip all gradient values to between $[-1, 1]$ and we finetune for at most 4000 iterations, or until we reach a nan gradient which we handle by safely exiting. We report the epochs with the highest PSNR and SSIM and we select that model for evaluating performance on all datasets. For measuring the time and space complexity of our model, we calculate inference time in seconds and size of model parameters in MB. Our code is written with PyTorch Paszke et al. (2019) and runs for at most 4 hours on one NVIDIA RTX 6000 48G GPU. The anonymous open source code for this paper along with instructions for reproducability can be found here.

## 7.5 RESULTS

We also give the performance of our method on the SwinIR-light ×2 and ×3 models for all quantization bitwidths.

| Method (×2) | Bit | Set5 | | Set14 | | B100 | | Urban100 | | Manga109 | |
|---|---|---|---|---|---|---|---|---|---|---|---|
| | | PSNR | SSIM | PSNR | SSIM | PSNR | SSIM | PSNR | SSIM | PSNR | SSIM |
| SwinIR-light | 32 | 38.15 | 0.9611 | 33.86 | 0.9206 | 32.31 | 0.9012 | 32.76 | 0.9340 | 39.11 | 0.9781 |
| Bicubic | 32 | 32.25 | 0.9118 | 29.25 | 0.8406 | 28.68 | 0.8104 | 25.96 | 0.8088 | 29.17 | 0.9128 |
| PTQ4ViT | 4 | 37.43 | 0.9571 | 33.19 | 0.9139 | 31.84 | 0.8950 | 31.54 | 0.9212 | 37.59 | 0.9735 |
| NoisyQuant | 4 | 37.50 | 0.9570 | 33.06 | 0.9101 | 31.73 | 0.8936 | 31.31 | 0.9181 | 37.47 | 0.9723 |
| 2DQuant | 4 | 37.87 | 0.9594 | 33.41 | 0.9161 | 32.02 | 0.8971 | 31.84 | 0.9251 | 38.31 | 0.9761 |
| CondiQuant | 4 | 38.03 | 0.9605 | 33.50 | 0.9180 | 32.16 | 0.8993 | 32.03 | 0.9282 | 38.57 | 0.9769 |
| CompSRT (ours) | 4 | **38.13** | **0.9610** | **33.81** | **0.9203** | **32.28** | **0.9009** | **32.57** | **0.9325** | **38.98** | **0.9778** |
| PTQ4ViT | 3 | 36.49 | 0.9510 | 32.49 | 0.9045 | 31.27 | 0.8854 | 30.16 | 0.9027 | 36.41 | 0.9673 |
| NoisyQuant | 3 | 35.32 | 0.9334 | 31.88 | 0.8911 | 30.73 | 0.8710 | 29.28 | 0.8835 | 35.30 | 0.9537 |
| 2DQuant | 3 | 37.32 | 0.9567 | 32.35 | 0.9106 | 31.60 | 0.8911 | 30.45 | 0.9086 | 37.24 | 0.9722 |
| CondiQuant | 3 | 37.77 | 0.9594 | 33.21 | 0.9151 | 31.94 | 0.8966 | 31.18 | 0.9197 | 38.01 | 0.9755 |
| CompSRT (ours) | 3 | **38.11** | **0.9609** | **33.82** | **0.9202** | **32.27** | **0.9008** | **32.53** | **0.9321** | **38.90** | **0.9775** |
| PTQ4ViT | 2 | 33.25 | 0.8923 | 30.22 | 0.8402 | 29.21 | 0.8066 | 27.31 | 0.8111 | 32.75 | 0.9093 |
| NoisyQuant | 2 | 30.13 | 0.7620 | 28.80 | 0.7536 | 28.26 | 0.7421 | 26.68 | 0.7627 | 30.40 | 0.8204 |
| 2DQuant | 2 | 36.00 | 0.9497 | 31.98 | 0.9012 | 30.91 | 0.8810 | 28.62 | 0.8819 | 34.40 | 0.9602 |
| CondiQuant | 2 | 37.15 | 0.9567 | 32.74 | 0.9103 | 31.55 | 0.8912 | 29.96 | 0.9047 | 36.63 | 0.9713 |
| CompSRT (ours) | 2 | **38.03** | **0.9605** | **33.70** | **0.9194** | **32.19** | **0.9294** | **32.22** | **0.9294** | **38.69** | **0.9770** |

Table 9: Performance comparison with SOTA methods for scale factor (×2) across different bit widths. All comparative results are taken from SwinIR-light Liang et al. (2021), PTQ4VIT Yuan et al. (2024), NoisyQuant Liu et al. (2023), 2DQuant Liu et al. (2024a), and CondiQuant Liu et al. (2025). Our method achieves superior performance across all datasets and bitwidths.

| Method (×3) | Bit | Set5 | | Set14 | | B100 | | Urban100 | | Manga109 | |
|---|---|---|---|---|---|---|---|---|---|---|---|
| | | PSNR | SSIM | PSNR | SSIM | PSNR | SSIM | PSNR | SSIM | PSNR | SSIM |
| SwinIR-light | 32 | 34.63 | 0.9290 | 30.54 | 0.8464 | 29.20 | 0.8082 | 28.66 | 0.8624 | 33.99 | 0.9478 |
| Bicubic | 32 | 29.54 | 0.8516 | 27.04 | 0.7551 | 26.78 | 0.7187 | 24.00 | 0.7144 | 26.16 | 0.8384 |
| PTQ4ViT | 4 | 33.77 | 0.9201 | 29.75 | 0.8272 | 28.62 | 0.7942 | 27.43 | 0.8361 | 32.50 | 0.9360 |
| NoisyQuant | 4 | 33.13 | 0.9122 | 29.06 | 0.8093 | 27.93 | 0.7754 | 26.66 | 0.8143 | 31.94 | 0.9293 |
| 2DQuant | 4 | 34.06 | 0.9231 | 30.12 | 0.8374 | 28.89 | 0.7988 | 27.69 | 0.8405 | 32.88 | 0.9389 |
| CondiQuant | 4 | 34.32 | 0.9260 | 30.29 | 0.8417 | 29.05 | 0.8039 | 28.05 | 0.8506 | 33.23 | 0.9431 |
| CompSRT (ours) | 4 | **34.56** | **0.9284** | **30.49** | **0.8454** | **29.17** | **0.8075** | **28.50** | **0.8598** | **33.83** | **0.9467** |
| PTQ4ViT | 3 | 32.75 | 0.9028 | 29.14 | 0.8113 | 28.06 | 0.7712 | 26.43 | 0.8014 | 31.20 | 0.9131 |
| NoisyQuant | 3 | 30.78 | 0.8511 | 27.94 | 0.7624 | 26.98 | 0.7153 | 25.43 | 0.7481 | 29.64 | 0.8792 |
| 2DQuant | 3 | 33.24 | 0.9135 | 29.56 | 0.8255 | 28.50 | 0.7873 | 26.65 | 0.8116 | 31.46 | 0.9235 |
| CondiQuant | 3 | 33.92 | 0.9224 | 30.02 | 0.8367 | 28.84 | 0.7986 | 27.37 | 0.8356 | 32.48 | 0.9367 |
| CompSRT (ours) | 3 | **34.54** | **0.9281** | **30.48** | **0.8451** | **29.16** | **0.8070** | **28.47** | **0.8589** | **33.79** | **0.9465** |
| PTQ4ViT | 2 | 29.96 | 0.7901 | 27.36 | 0.7030 | 26.74 | 0.6590 | 24.56 | 0.6460 | 27.37 | 0.7390 |
| NoisyQuant | 2 | 27.53 | 0.6641 | 25.77 | 0.5952 | 25.37 | 0.5613 | 23.59 | 0.5739 | 26.03 | 0.6632 |
| 2DQuant | 2 | 31.62 | 0.8887 | 28.54 | 0.8038 | 27.85 | 0.7679 | 25.30 | 0.7685 | 28.46 | 0.8814 |
| CondiQuant | 2 | 33.00 | 0.9130 | 29.44 | 0.8253 | 28.45 | 0.7882 | 26.36 | 0.8080 | 30.88 | 0.9203 |
| CompSRT (ours) | 2 | **34.17** | **0.9248** | **30.21** | **0.8401** | **28.97** | **0.8017** | **27.86** | **0.8456** | **33.11** | **0.9414** |

Table 10: Performance comparison with SOTA methods for scale factor (×3) across different bit widths. All comparative results are taken from SwinIR-light Liang et al. (2021), PTQ4VIT Yuan et al. (2024), NoisyQuant Liu et al. (2023), 2DQuant Liu et al. (2024a), and CondiQuant Liu et al. (2025). Our method achieves superior performance across all datasets and bitwidths.

## 7.6 COMPLEXITY FOR ALL MODELS

For scaling factors ×2, ×3, ×4, we include the time required to complete a single forward pass on our 2-4 bit quantized model and our 3-4 bit quantized and unstructured pruned model on an image from the Set5 benchmark dataset in seconds. We also measured the size of the model's trainable parameters in megabytes (MB) and in bits per parameter $b$, comparing with 2DQuant. The results are included in Table 11. We do not perform pruning for the 2 bit quantized models, as considering the 1-bit mask, this does not give storage gains. This table complements Table 4, by providing more results. We can see that the inference times and MB of parameters follow a similar trend to the ×4 bit 4 models in Table 4. We have roughly 0.04 seconds of additional inference time across all models compared with 2DQuant due to the effect of the Hadamard transform.

| Scale ×2 | | | | | | Scale ×3 | | | | | |
|---|---|---|---|---|---|---|---|---|---|---|---|
| Method | Bit | Pruning | $s$ | MB | $b$ | Method | Bit | Pruning | $s$ | MB | $b$ |
| 2DQuant | 4 | No | 0.040 | 13.92 | 4 | 2DQuant | 4 | No | 0.042 | 13.95 | 4 |
| Ours | 4 | No | 0.084 | 13.92 | 4 | Ours | 4 | No | 0.084 | 13.95 | 4 |
| Ours | 4 | Yes | 0.084 | 13.87 | 3.4 | Ours | 4 | Yes | 0.084 | 13.90 | 3.4 |
| 2DQuant | 3 | No | 0.040 | 13.84 | 3 | 2DQuant | 3 | No | 0.042 | 13.87 | 3 |
| Ours | 3 | No | 0.084 | 13.84 | 3 | Ours | 3 | No | 0.084 | 13.87 | 3 |
| Ours | 3 | Yes | 0.084 | 13.82 | 2.8 | Ours | 3 | Yes | 0.084 | 13.85 | 2.8 |
| 2DQuant | 2 | No | 0.039 | 13.75 | 2 | 2DQuant | 2 | No | 0.042 | 13.79 | 2 |
| Ours | 2 | No | 0.086 | 13.76 | 2 | Ours | 2 | No | 0.084 | 13.79 | 2 |

| Scale ×4 | | | | | |
|---|---|---|---|---|---|
| Method | Bit | Pruning | $s$ | MB | $b$ |
| 2DQuant | 3 | No | 0.041 | 13.91 | 3 |
| Ours | 3 | No | 0.0883 | 13.91 | 3 |
| Ours | 3 | Yes | 0.0883 | 13.90 | 2.8 |
| 2DQuant | 2 | No | 0.041 | 13.83 | 2 |
| Ours | 2 | No | 0.090 | 13.83 | 2 |

Table 11: Comprehensive evaluation of inference time in seconds ($s$), MB of parameters, and effective bits-per-parameter $b$ across scale factors ×2 and ×3, and ×4 for bit widths 2-4 comparing with 2DQuant. Pruning indicates whether there is pruning being performed.

## 7.7 MAMBAIRV2-LIGHT RESULTS

To showcase the generalizability of our method, we also extend our quantization framework to quantize the attention layers, linear layers and batch matrix multiplication weights and activations of

MambaIRv2-light Guo et al. (2024). We measure our performance for the extreme 2 bit quantization case for MambaIRv2-light $\times 2$, $\times 3$ and $\times 4$ models on all of our five datasets. The results are given in Table 12. We see that our method performs well, and we greatly close the gap between the FP Mamba model and the 2-bit quantized model, although we do not quantize convolutional layers. Furthermore, we are the first work to quantize MambaIRv2-light.

| Method | Bit | Set5 | | Set14 | | B100 | | Urban100 | | Manga109 | |
|--------|-----|------|------|-------|------|------|------|----------|------|----------|------|
| | | PSNR | SSIM | PSNR | SSIM | PSNR | SSIM | PSNR | SSIM | PSNR | SSIM |
| MambaIRv2-light ($\times 2$) | 32 | 38.26 | 0.9615 | 34.09 | 0.9221 | 32.36 | 0.9019 | 33.26 | 0.9378 | 39.35 | 0.9785 |
| Ours ($\times 2$) | 2 | 38.15 | 0.9611 | 33.90 | 0.9208 | 32.29 | 0.9010 | 32.80 | 0.93 | 39.05 | 0.9779 |
| MambaIRv2-light ($\times 3$) | 32 | 34.71 | 0.9298 | 30.68 | 0.8483 | 29.26 | 0.8098 | 29.01 | 0.8689 | 34.41 | 0.9497 |
| Ours ($\times 3$) | 2 | 34.40 | 0.9268 | 30.36 | 0.8426 | 29.08 | 0.8056 | 28.29 | 0.8565 | 33.51 | 0.9440 |
| MambaIRv2-light ($\times 4$) | 32 | 32.51 | 0.8992 | 28.84 | 0.7878 | 27.75 | 0.7426 | 26.82 | 0.8079 | 31.24 | 0.9182 |
| Ours ($\times 4$) | 2 | 32.25 | 0.8960 | 28.70 | 0.7844 | 27.62 | 0.7388 | 26.37 | 0.7965 | 30.75 | 0.9126 |

Table 12: Performance comparison with MambaIRv2-light Guo et al. (2024) for the 2-bit extreme quantization case. Our method greatly closes the gap between the quantized and FP models.

## 7.8 LLM USAGE

We used LLMs to assist—but not replace—our research workflow. Specifically, LLMs were employed to (i) help draft and refactor code snippets and experimental scripts, (ii) brainstorm and clarify ideas and concepts discussed in the paper, (iii) suggest edits and critiques on early drafts, and (iv) provide limited writing assistance for grammar and phrasing. All model outputs were reviewed, verified, and, where needed, rewritten by the authors; we independently implemented, tested, and validated every algorithmic choice and experimental result reported. No proprietary, confidential, or unreleased data were provided to the models. LLMs were not used to generate or fabricate data, analyses, or citations, and they are not listed as authors. The authors bear full responsibility for the paper's content, including the correctness of code, experiments, and references.

