# OpenReview forum: "CompSRT: Quantization and Pruning for Image Super Resolution Transformers"
_ICLR.cc/2026/Conference — Submitted to ICLR 2026_

### Official Review · Reviewer_9peg · 2025-10-21

**Soundness:** 3
**Presentation:** 3
**Contribution:** 2
**Rating:** 4
**Confidence:** 4

**Summary:**

This paper proposes two techniques, i.e., Hadamard transform before quantization and decomposition of quantization parameters. Compared to the state-of-the-art method, i.e., CondiQuant, the proposed method outperforms by a large margin, especially for low bit quantization cases.

**Strengths:**

This paper firstly shows that Hadamard transform is effective for Swin-IR in the SR domain.
The proposed decomposition is new and can reflect the actual quantization scale and bias as well as learnability depending on the distributions.
The proposed method outperforms SOTA by a large margin.

**Weaknesses:**

1. Hadamard transform is being widely used in may video coding and deep neural network compression (e.g., LLM, diffusion models). And the major performance gain of the proposed method comes from the Hadamard transform. Although This paper argues that Hadamard transform can concentrate energy to low frequency regions as make high frequency region sparse, this property of transforms (e.g., FFT, DCT, DST and DWHT) is well known and proved in many literature for natural images and W/A of deep neural networks [1-3]. This limits the novelty of this paper.

[1] Xinyu Wang, et al. 2025. HadaNorm: Normalization after Hadamard Incoherence for Activation Quantization. In Proceedings of the 42nd International Conference on Machine Learning (ICML 2025).

[2] Song Han, et al. 2024. QuIP#: Even Better LLM Quantization with Hadamard Incoherence and Lattice Codebooks. In Proceedings of the 41st International Conference on Machine Learning (ICML 2024).

[3] Tim Dettmers, et al. 2024. QuaRot: Outlier-Free 4-bit Inference in Rotated LLMs. In Advances in Neural Information Processing Systems (NeurIPS 2024).


2. the proposed scalar decomposition can be viewed as a variant of LSQ which learns quantization parameters. To verify the effectiveness of the proposed hybrid (e.g., a uniform quant-based fixed scale with a learnable variable) method should be compared with LSQ.

3. Nowdays, there are a lot of SR models based on diffusion models, Mamba [4, 5] etc.. This paper improves the performance of quantized Swin-IR models only. This makes the impact of this paper limited.

[3] Hang Guo, Jinmin Li, Tao Dai, Zhihao Ouyang, Xudong Ren, and Shu-Tao Xia. Mambair: A simple baseline for image restoration with state-space model. In European conference on computer vision, pp. 222–241. Springer, 2024a.  Xiangyu Guo, Kai Zhang, Jingyun

[4] Liang, Yulun Wang, and Radu Timofte. Mambairv2: Hybrid mamba-transformer architecture for efficient image restoration. arXiv preprint arXiv:2501.01234, 2025.

**Questions:**

Please read the Weakness section and answer the questions.

**Details Of Ethics Concerns:**

No ethical concerns were found in the paper.

---

> ### Author Response · Authors · 2025-11-21
>
> Hello,
> We thank you for taking the time to read our paper and provide feedback. To answer your concerns:
>
> Behavior of Hadamard transform:  there have been many attempts to explain the behavior of the Hadamard transformation, with quip and quip# explaining it via theorems about incoherence processing, Hadanorm explaining it through an argument centering on the central limit theorem and spinquant stated that random rotations like the Hadamard blend large and small weights together into a well-behaved distribution. However, in the literature on quantization there has not yet been an empirical analysis using the values of the activations and weights along with statistical tests to explain how the transform functions. To better convey this point, the line “Therefore, the mechanism for the Hadamard’s lowering of the errors remains unexplored” now reads:
> “Previous arguments on how the transform functions hinges on several concepts: incoherence processing, the central limit theorem and channel mixing. Tseng et al. (2024) found theorems that Hadamard transformations concentrate the entry magnitudes of distributions in a process called incoherence processing, and Liu et al. (2024b) stated that random rotations blend large and small weights together into a well-behaved distribution. Federici et al. (2025) cited the central limit theorem as to why the distributions after the Hadamard transform tend towards gaussian and thus have outliers reduced. However, an empirical analysis using the values of activations and weights and statistical analysis on how the transform functions in the quantization domain still remains underexplored. Furthermore, combining quantization with other compression techniques in image super resolution transformers remains unexplored.”
> To reiterate, we are the first work to analyze the distributions of weights and activations empirically with tests of statistical significance to reason about the behavior of the Hadamard Transformation.
>
> Difference from LSQ:  Our work differs from LSQ in that we are not actually learning the quantization scalar and zero points. The quantization scalar is determined by learned upper and lower bounds of the quantization range, as put forth by 2DQuant and the integer zero point is the learned lower bound of the quantization range. However, we are learning two additional parameters that modify the scalar and the integer zero point, and these two additional parameters have not been introduced in the literature. To quantify the effect of only our parameters, we run an ablation where we do not learn these upper and lower bounds and only learn our parameters on the 2-bit x2 model. The results are given in Table 7 in the supplementary materials. From the ablations Figure 6, we see that the Hadamard transforms on their own do not surpass the performance of CondiQuant. However, in table 7 we see that only learning our parameters and not learning the upper and lower bounds improves performance, surpassing 2Dquant, but not surpassing CondiQuant, showing that our parameters are uniquely beneficial. However, they must be combined with learning the upper and lower bounds of the clipping range, which is not learning the quantization scalar or zero points, but rather learning the building blocks that create the scalar and zero point to have the combined effect of surpassing CondiQuant and achieving the SOTA performance.
>
> Different backbones:  Thank you for your suggestion. We now benchmark our quantization method on MambaIRv2-light. Please see the full results in the supplementary materials (section 7.7). We are also the first work to quantize MambaIRv2-light and our method greatly closes the gap between the FP and quantized model.

---

> ### Author Response · Authors · 2025-11-24
>
> Hello,
>
> We believe our rebuttal fixed concerns, and due to the coming deadline for the end of the discussion, we kindly remind the reviewer of the due date. We are happy to address further questions. Please let us know.
>
> Thank you

---

> ### Author Response · Authors · 2025-12-03
>
> Dear Area Chairs,
>
> Thank you so much for taking the time to read this and considering our paper. We would like to provide a brief, high-level summary of our contributions and clarify how we have addressed this reviewer's concerns during the rebuttal period.
>
> This reviewer raised an issue with the contribution of our Hadamard analysis, and we reiterate that our work is the only work to study with tests of statistical significance such as the wilcoxon signed rank test what changes the Hadamard transform induces and whether these changes in activations and weights are statistically significant and with what effect size, while prior work have only posited theories or empirical analysis only on activations with no tests of significance or plots of activations and weights pre and post transformation, again without any statistical significance analysis. We believe that we have pinned down the exact mechanisms for the Hadamard's effectiveness and show that these properties statistically significantly happen for both activations and weights, which is a more in depth and novel contribution in this space.
>
> Furthermore, this reviewer raised an issue regarding the novelty of our two additional parameters which modify the quantization scalar and zero points. We push back against this by first stating that we do not learn the quantization scalar and zero point directly like in LSQ/LSQ+, and instead we only learn the upper and lower bounds of the clipping range (which was the main contribution of 2DQuant), in addition to our parameters. We also show with the ablation study on the 2-bit quantized SwinIr-light $\\times 2$ model given in above comments that the two additional parameters we added, which modify the scalar after it has been calculated from the upper and lower bounds and the zero point which is instantiated as the lower bound, do not in fact converge to learning the scalar and zero points directly.  From that ablation we see that only learning our parameters surpasses the performance of 2DQuant, but if our parameters converged to learning the scalar then their performance would not surpass 2DQuant. We also see that our approach must be combined with learning the upper and lower bounds of the clipping range to surpass CondiQuant and achieve SOTA, showing that both our parameters and the upper and lower bounds uniquely give their own benefits, and do not converge to the same behavior. This table is provided in the supplementary materials as well.
>
> Regarding other backbones, we followed the suggestion of this reviewer to extend our framework to MambaIRv2-light, and provided the results in the above comments as well. We reiterate that we are also the first work to quantize MambaIRv2-light. Overall, we addressed every doubt raised by this reviewer, and we hope that you can consider our additional experiments and statistical analysis when making your decision.
>
> Thank you again.

---

### Official Review · Reviewer_VCpf · 2025-10-31

**Soundness:** 3
**Presentation:** 3
**Contribution:** 2
**Rating:** 4
**Confidence:** 4

**Summary:**

The paper targets compressing SwinIR-light for image super-resolution via a Hadamard-guided PTQ pipeline, a lightweight reparameterization of the quantization scale and zero-point, and 40% weight pruning at 3–4 bits. The key empirical claim is that, for SwinIR-light, Hadamard transforms do not flatten distributions but do shrink ranges and increase near-zero mass, which reduces quantization error. Extensive benchmarks show consistent PSNR/SSIM gains over previous methods. With pruning, bits-per-parameter drop by 6.67–15% at comparable quality.

**Strengths:**

1. The paper provides a fairly comprehensive validation within SwinIR-light of the mechanism by which Hadamard transforms are effective.
2. By decoupling the quantization step size and zero-point into two learnable parameters, the method intuitively increases representational and optimization flexibility while remaining simple to implement and directly pluggable into existing PTQ pipelines.
3. An integrated practice combining pruning and quantization is presented.

**Weaknesses:**

1. The claim that “This operation has been said to flatten the matrices by distributing the magnitude of outliers and that is how the errors get reduced, but the exact mechanism has not been explored nor has the flatness or normality of distributions been tested.” is not fully accurate. QuaRot [A] demonstrates end-to-end 4-bit LLM inference using Hadamard-based rotations to mitigate outliers, with distributional visualizations supporting the “flattening” effect. SpinQuant [B] shows the mechanism is the rotation itself—random rotations vary widely, while learned rotations (via Cayley optimization) yield larger gains, directly explaining why errors decrease. HadaNorm [C] further analyzes when Hadamard mixing is weakened by per-channel mean/scale, and proposes centering+rescaling before Hadamard with quantitative improvements—a more systematic treatment of activation distributions. In light of these results, the paper’s statement is incomplete, and the corresponding contribution is consequently weakened.
References:
[A] Ashkboos et al., QuaRot: Outlier-Free 4-Bit Inference in Rotated LLMs.
[B] Liu et al., SpinQuant: LLM Quantization with Learned Rotations.
[C] Federici et al., HadaNorm: Diffusion Transformer Quantization through Mean-Centered Transformations.
2. The paper lacks evaluations on real-world degradation benchmarks to verify robustness under non-ideal distribution shifts.
3. Experiments are currently limited to SwinIR-light. The paper lacks experiments on additional Transformer-based backbones to demonstrate architecture-agnostic effectiveness.
4. The paper lacks inference throughput results across varying input resolutions and multiple upscaling factors. Please report latency or FPS under representative settings.

**Questions:**

1. Please further explain why, under the 3-bit setting, modest gains appear in the 10–30% range. What drives this phenomenon? Does it persist across repeated runs with different random seeds?
2. The paper measures the effect of the Hadamard transform via the zero-neighborhood ratio with the threshold ε fixed at 0.05. How sensitive are your results to ε?
3. Why do the results in Table 3 show that the inference time and number of parameters for the pruned model do not decrease in roughly proportional terms?

---

> ### Author Response · Authors · 2025-11-21
>
> Hello,
> We thank you for taking the time to read our paper and provide feedback. To answer your concerns:
>
> Hadamard matrix: We have clarified the discussion of prior work and replaced the original passage with: “Previous literature on quantization of LLMs (Ashkboos et al. (2024), Liu et al. (2024b), Federici et al. (2025), Chee et al. (2023), Sun et al. (2024), Tseng et al. (2024)) has found that outliers in weights and activations cause performance degradations. Sun et al. (2024) dubbed the removal of outliers as increasing the “flatness” of weights and activations and Federici et al. (2025), and Tseng et al. (2024) stated that Hadamard transformations can increase flatness of distributions.” To contextualize the role of prior explanations, we note that quip/quip# emphasize incoherence processing, HadaNorm uses the central limit theorem, and SpinQuant argues that random rotations blend large and small weights. However, quantization-specific empirical analysis has been lacking. Thus, we revised: “Therefore, the mechanism for the Hadamard’s lowering of the errors remains unexplored.” to: “Previous arguments on how the transform functions hinge on several concepts: incoherence processing, the central limit theorem and channel mixing. Tseng et al. (2024) found theorems that Hadamard transformations concentrate the entry magnitudes of distributions in a process called incoherence processing, and Liu et al. (2024b) stated that random rotations blend large and small weights together into a well-behaved distribution. Federici et al. (2025) cited the central limit theorem as to why the distributions after the Hadamard transform tend towards gaussian and thus have outliers reduced. However,  a detailed empirical study of both activations and weights, including statistical significance tests of how these transforms affect outliers still remains
> underexplored. Furthermore, combining quantization with other compression techniques in image super resolution transformers remains unexplored.”
>
> Real world datasets: Our experiments use the five standard SR benchmarks; four (Set5, Set14, B100, Urban100) contain real-world images, so our evaluation already occurs on real-world data.
>
> Other backbones: Now, we also benchmark our method on MambaIRv2-light. Full results are given in supplementary materials (section 7.7). We are also the first work to quantize MambaIRv2-light and our method greatly closes the gap between the FP and quantized model, showing generalizability.
>
> Comprehensive inference profiling: Table 4 reports end-to-end latency for the x4 4-bit quantized and pruned model. We now additionally include full inference-time results across all scale factors and bit-widths in the supplementary (Table 11).
>
> To answer your questions:
> 3-bit quantization (8 levels) reduces quantization error by ~4× relative to 2-bit (4 levels), placing 3-bit above the “critical resolution threshold” that preserves layer behavior. In contrast, 2-bit is an extreme low-bit regime with severe dynamic-range limitations and high quantization noise. Hence 2-bit often requires substantial engineering, while 3-bit maintains representational structure and remains closer to full-precision. This behavior persists across random seeds.
>
> Epsilon-sensitivity:  The epsilon threshold is diagnostic rather than algorithmic: it allows us to examine whether the Hadamard transform increases the proportion of values near zero, enabling pruning or lower-error quantization. We sweep epsilon from [0.01, 1] in 0.1 increments and report results in Table 6 of the supplementary. As epsilon increases, p-values grow and effect sizes shrink, consistent with distributions concentrated near zero with small standard deviations, but many results remain statistically significant, indicating low sensitivity to epsilon.
>
> Proportional decrease: Parameter counts in Table 4 do not scale directly with the nominal 40% pruning ratio because we retain a binary mask of the same shape to record pruned vs. active weights. Furthermore, we only prune quantized weights, which do not account for the convolution layers, which remain unquantized and unpruned. This simple mechanism demonstrates feasibility for SwinIR, though more advanced schemes could yield larger reductions. The final footprint includes both surviving weights and the 1-bit mask, reducing but not eliminating storage. Similarly, inference time does not scale proportionally: kernels must still access and apply the mask, and because we use fake quantization, runtime benefits are limited compared to true low-bit kernels that could exploit the effective 3.4 bits/parameter (e.g., through low-bit GEMM).

---

> ### Author Response · Authors · 2025-11-24
>
> Hello,
>
> We believe our rebuttal fixed concerns, and due to the coming deadline for the end of the discussion, we kindly remind the reviewer of the due date. We are happy to address further questions. Please let us know.
>
> Thank you

---

> > ### Comment · Reviewer_VCpf · 2025-11-27
> >
> > The athours' response partially addresses my concerns. Inaccurate literature reviews and insufficient comparative experiments limit the impact of the paper. I keep my rating.

---

> ### Author Response · Authors · 2025-11-27
>
> Can you elaborate more on the inaccurate literature review? What part of our literature review is inaccurate? Furthermore, what part of our comparative experiments is insufficient? We have carefully revised the literature reviews and we believe them to be accurate. We have also fully benchmarked our method on SwinIR-light across all bitwidths and scale factors and extended our method to MambaIRv2-light under the extreme compression case (2-bit) for all scale factors to showcase generalizability.

---

> > ### Author Response · Authors · 2025-11-30
> >
> > To ensure our claims are as accurate and fair as possible, we have refined our literature review. Previously, when citing SpinQuant (Liu et al., 2024b), we did not explicitly acknowledge that they evaluate the kurtosis of activation distributions to show that rotations reduce outliers. We originally wrote: "Liu et al. (2024b) stated that random rotations blend large and small weights together into a well- behaved distribution. Federici et al. (2025) cited the central limit theorem as to why the distributions after the Hadamard transform tend towards gaussian and thus have outliers reduced. However, an empirical analysis using the values of activations and weights and statistical analysis on how the transform functions in the quantization domain still remains underexplored." We have revised this passage to read:
> > "Liu et al. (2024b) stated that random rotations blend large and small weights together into a well-behaved
> > distribution and empirically analyzed activation distributions via their kurtosis. Federici et al. (2025)
> > cited the central limit theorem as to why the distributions after the Hadamard transform tend towards
> > Gaussian and have outliers reduced. However, a detailed empirical study of both activations and
> > weights, including statistical significance tests of how these transforms affect outliers still remains
> > underexplored."  In our revision, we clarify that SpinQuant analyzes activation distributions via kurtosis, while highlighting that our work goes further by performing statistical significance testing on both activations and weights, providing a more in-depth analysis of how Hadamard transforms reduce outliers.

---

> > > ### Author Response · Authors · 2025-12-03
> > >
> > > Dear Area Chairs,
> > >
> > > Thank you so much for taking the time to read this and considering our paper. We would like to provide a brief, high-level summary of our contributions and clarify how we have addressed this reviewer's concerns during the rebuttal period.
> > >
> > > This reviewer raised an issue with the contribution of our Hadamard analysis, and we reiterate that our work is the only work to study with tests of statistical significance such as the wilcoxon signed rank test what changes the Hadamard transform induces and whether these changes in activations and weights are statistically significant and with what effect size, while prior work have only posited theories or empirical analysis only on activations with no tests of significance or plots of activations and weights pre and post transformation, again without any statistical significance analysis. We believe that we have pinned down the exact mechanisms for the Hadamard's effectiveness and show that these properties statistically significantly happen for both activations and weights, which is a more in depth and novel contribution in this space.
> > >
> > > Furthermore, this reviewer raised an issue with insufficient experiments, we which addressed by including comprehensive end to end inference and size profiling for all models, rigorous ablation study of the effect of our parameters, extension to another backbone and study of epsilon sensitivity, which are all provided as tables in the above comments for your reference, and we included these additional results in the supplementary. Furthermore, we also show further compression in pruning by exploring the structured channel wise pruning and we give the results in the above comments as well. Regarding the literature review, we have revised it to include every additional source provided by the reviewers and a revision of our claims to show our novelty and contribution in using statistical significance analysis. While the reviewer did not get to specify what part of our experiments and literature review remain inaccurate, we hope that you can consider our efforts in providing the additional experiments shown above and our refinement of our claims.
> > >
> > > Thank you so much for your time and consideration.

---

### Official Review · Reviewer_Vw3V · 2025-11-03

**Soundness:** 3
**Presentation:** 3
**Contribution:** 3
**Rating:** 6
**Confidence:** 3

**Summary:**

This paper proposes CompSRT, a post‑training compression pipeline for the SwinIR‑light super‑resolution model that combines (i) Hadamard‑guided quantization of weights and activations and (ii) scalar decomposition of the quantization scale and zero offset.

A central empirical claim is that Hadamard transforms do not “flatten” weight/activation distributions in SwinIR‑light; instead, they reduce value ranges and increase mass near zero, which the authors argue lowers quantization error.

Quantization is applied to attention/MLP components, and the method is optionally combined with 40% unstructured weight pruning. CompSRT outperforms prior PTQ baselines on five standard benchmarks. The authors also report a small per‑image runtime overhead from the extra Hadamard operations and a reduction in “bits per parameter” when pruning.

**Strengths:**

1) The paper provides consistent PSNR/SSIM improvements across five datasets and three scales at 2–4 bits.

2) The experimental evaluation is very detailed and the results are supported by various statistical tests. The authors demonstrated that the Hadamard transformations reduce quantization errors in matrices by reducing the ranges of the values, and concentrating values around 0.
The paper goes beyond intuition, using paired tests and effect sizes to argue the Hadamard's benefit arises from range shrinkage and increased near‑zero mass.

3) The text is well writted; where the transforms are inserted and how scalar decomposition is optimized are clearly described.

4) The technique is lightweight conceptually, slots into PTQ, and composes with pruning (40%) to reduce effective bits per parameter.

5) The authors provided good ablation study. In particular, the study indicates that scalar decomposition contributes strongly; Hadamard on both weights and activations further helps, aligning with the distributional evidence.

6) The paper includes a reproducibility statement and an anonymous code link.

**Weaknesses:**

1) Because CondiQuant code is unavailable, qualitative results use 2DQuant; this limits the strength of the visual SOTA claim. A controlled, code‑level visual comparison with CondiQuant (if possible) or an alternative strong visual baseline would help.

2) The paper reports a per‑image overhead due to extra FP Hadamard operations. There is no end‑to‑end profiling (e.g., batch throughput, latency on edge hardware, kernel fusion feasibility).

3) The 1‑bit dense mask plus storing the pruned tensor is simple, but practical implementations often require not standard sparse formats.

4) Results focus on SwinIR‑light; it is unclear whether the same gains hold for larger SwinIR variants or different SR architectures (EDSR/RDN) and for different activation/weight quantization granularities. The method likely generalizes, but evidence is limited to one backbone

5) The signed‑rank tests for superiority vs. CondiQuant are conducted over only five datasets per configuration.

6) Robustness to \epsilon (set to 0.05) and to other orthonormal transforms (e.g., DCT, random orthogonal) is not explored, so it remains uncertain whether Hadamard is uniquely effective.

**Questions:**

1) Figures 3–4 show both Hadamard and inverse‑Hadamard nodes. Are these executed during inference for every forward pass? If so, what is the measured layer‑wise cost on typical hardware

2) How does your calibration set size and finetuning budget (up to 4k iters with Adam) compare to those used by 2DQuant/CondiQuant? Could differences partly explain the observed gains? Please clarify the exact data used during finetuning and whether early stopping was based on the validation set.

3) How sensitive are the range‑reduction and near‑zero‑mass effects (and resulting PSNR) to the \epsilon threshold and to the choice of orthonormal transform (Hadamard vs. DCT vs. random orthogonal)? A small ablation would clarify whether the effect is Hadamard‑specific.

4) What granularity is used (per‑tensor vs. per‑channel/group) for activations and weights? Would scalar decomposition still help under finer granularity?

5) Have you tried the method on SwinIR‑base or on CNN SR backbones (EDSR/RDN)? Even a subset of experiments would bolster generality claims.

6) If CondiQuant visuals are unavailable, could you at least add more visual crops against a strong available baseline (e.g., 2DQuant with equal finetuning budget) on the same images and provide details of the experiment to ensure reproducibility?

---

> ### Author Response · Authors · 2025-11-21
>
> Hello,
> We thank you for taking the time to read our paper and provide feedback. To answer your concerns:
>
> Condiquant unavailability: Because Condiquant does not have released code, we compare our results with the previous SOTA, 2DQuant. To be clear, 2DQuant is the SOTA method for SwinIR prior to 2DQuant. Furthermore, we want to stress that we have the exact same finetuning budget and procedure as 2DQuant, as we follow 2DQuant and our codebase is built on top of theirs. However, the 2DQuant paper and code are not aligned: in the 2DQuant paper, they cite that they have used 3000 iterations, but in their code, it shows 4000 iterations. We use these 4000 iterations as stated in their codebase. Therefore, we do not finetune for more iterations or with more data examples than them and thus our results are comparable.
>
> Other backbones: To show generalizability, now we also benchmark our quantization method on MambaIRv2-light. Full results are given in supplementary materials (section 7.7). We are also the first work to quantize MambaIRv2-light and our method greatly closes the gap between the FP and quantized model.
>
> 1-bit dense mask: Yes, thank you for raising this great point. We were more interested in exploring the possibility of performing pruning and quantization, and aiming to get a theoretical understanding of the performance. Now we also benchmark performing structured pruning of channels with the lowest average absolute magnitudes and see that this structured pruning results in worse performance degradations in Table 8 in the supplementary.
>
> End-to-end profiling:   We provided end to end profiling by providing the time it takes to complete a forward pass on one image from set5 for the x4 4 bit quantized and pruned models in Table 4 in our paper. Now, we also provide the inference time in seconds for all scale factors and bitwidths in the supplementary materials Table 11.
>
> signed‑rank tests for superiority:    The signed‑rank tests for superiority vs. CondiQuant are conducted over the 5 most commonly chosen datasets for evaluating the performance of SR models.
>
> To answer your specific questions:
> Figures 3-4: These are executed for every forward pass, and the inference time in seconds for the 4 bit x4 model is given in Table 4 and for the other configurations in table 11 in the supplementary materials. In the x4 case, we see that the Hadamard transformation increases the time taken for a forward pass by 0.048 seconds.
> Finetuning budget: We have the same finetuning budget and dataset used as 2DQuant. CondiQuant only performs condition number calibration and thus their budget is not directly comparable to ours. Furthermore, CondiQuant performs better than 2DQuant with a different calibration strategy, thus showing that the calibration strategy is not the main cause of the performance, as is the case in our model as well. The gains that we see cannot be attributed to just the finetuning strategy alone. Furthermore, there was no early stopping based on the validation set.
>
> Epsilon-sensitivity: To clarify, the epsilon threshold is a way to observe and reason about the distribution of the weights and activations, not an actual part of the algorithm that will affect downstream PSNR. We wanted to observe if there was an increase in the proportion of values around 0 after the Hadamard Transform, which would allow us to prune those values or quantize them with lower errors. Therefore, we chose a small epsilon to observe this effect. To see whether we can still observe a statistically significant increase in the proportion of values in the [-epsilon,epsilon] region as we vary the values of epsilon, we repeated our analysis with epsilon from [0.01,1], testing whether the proportion of values increased for each 0.1 increment. The results are included in table 6 in the supplementary.  As we increase the value of epsilon, we see larger p-values and smaller effect sizes, which makes sense as many of these matrices are distributed around 0 with small standard deviations. However, many p-values are statistically significant, making this analysis not very sensitive to epsilon.
>
> Granularity: We have chosen per tensor level granularity, and we believe that increasing granularity would add more quantization scalars and zero points which will improve the performance, but in our method, we only add two parameters per tensor, not many more which would be what would happen with increased granularity. For example with per channel granularity, we would add more than 2, as any weight or activation matrix will have more than 2 channels.Therefore, finer granularity will help on its own, but it will add more parameters than we are adding now.
>
> Visual examples of methods with equal finetuning budget:  All of our examples are against 2DQuant, the previous SOTA before CondiQuant, with equal finetuning budget. We also provide the details of the experiments in our github repository for exact reproducibility.

---

> ### Author Response · Authors · 2025-11-24
>
> Hello,
>
> We believe our rebuttal fixed concerns, and due to the coming deadline for the end of the discussion, we kindly remind the reviewer of the due date. We are happy to address further questions. Please let us know.
>
> Thank you

---

> > ### Author Response · Authors · 2025-12-03
> >
> > Dear Area Chairs,
> >
> > Thank you so much for taking the time to read this and considering our paper. We would like to provide a brief, high-level summary of our contributions and clarify how we have addressed this reviewer's concerns during the rebuttal period. This reviewer raised an issue with whether our finetuning budget is the same as 2Dquant, and we have stated explicitly that it is. Thus, our results are comparable with 2DQuant and any gains we have is not attributed to more finetuning.
> >
> > Furthermore, regarding other backbones, we now we also benchmark our quantization method on MambaIRv2-light. Full results are given below and in supplementary materials. We are the first work to quantize MambaIRv2-light.
> >
> > | Method | Bit | Set5 PSNR | Set5 SSIM | Set14 PSNR | Set14 SSIM | B100 PSNR | B100 SSIM | Urban100 PSNR | Urban100 SSIM | Manga109 PSNR | Manga109 SSIM |
> > |--------|-----|-----------|-----------|------------|------------|-----------|-----------|---------------|---------------|---------------|---------------|
> > | MambaIRv2-light ($\times 2$) | 32 | 38.26 | 0.9615 | 34.09 | 0.9221 | 32.36 | 0.9019 | 33.26 | 0.9378 | 39.35 | 0.9785 |
> > | Ours ($\times 2$)           | 2  | 38.15 | 0.9611 | 33.90 | 0.9208 | 32.29 | 0.9010 | 32.80 | 0.9300 | 39.05 | 0.9779 |
> > |                              |     |       |        |        |        |       |        |        |        |        |        |
> > | MambaIRv2-light ($\times 3$) | 32 | 34.71 | 0.9298 | 30.68 | 0.8483 | 29.26 | 0.8098 | 29.01 | 0.8689 | 34.41 | 0.9497 |
> > | Ours ($\times 3$)           | 2  | 34.40 | 0.9268 | 30.36 | 0.8426 | 29.08 | 0.8056 | 28.29 | 0.8565 | 33.51 | 0.9440 |
> > |                              |     |       |        |        |        |       |        |        |        |        |        |
> > | MambaIRv2-light ($\times 4$) | 32 | 32.51 | 0.8992 | 28.84 | 0.7878 | 27.75 | 0.7426 | 26.82 | 0.8079 | 31.24 | 0.9182 |
> > | Ours ($\times 4$)           | 2  | 32.25 | 0.8960 | 28.70 | 0.7844 | 27.62 | 0.7388 | 26.37 | 0.7965 | 30.75 | 0.9126 |
> >
> > Regarding the pruning, for more practicality, we now we also benchmark performing structured pruning of channels with the lowest average absolute magnitudes and see that this structured pruning results in worse performance degradations in Table 8 in the supplementary and also below.
> > | Method ($\times 4$) | Bit | $b$ | MB    | $s$   | PSNR  | SSIM   |
> > |---------------------|-----|----:|------:|------:|-------|--------|
> > | Ours                | 4   | 3.4 | 13.95 | 0.092 | 32.02 | 0.8900 |
> > | Ours                | 4   | 2.4 | 13.86 | 0.120 | 29.92 | 0.8471 |
> > | Ours                | 3   | 2.8 | 13.90 | 0.088 | 31.74 | 0.8900 |
> > | Ours                | 3   | 1.8 | 13.82 | 0.130 | 29.90 | 0.8462 |
> > | Ours                | 2   | 2.0 | 13.83 | 0.090 | 31.44 | 0.8820 |
> > | Ours                | 2   | 1.2 | 13.77 | 0.120 | 29.84 | 0.8460 |
> >
> > Regarding end to end profiling, we provided end to end profiling by providing the time it takes to complete a forward pass on one image from set5 for the x4 4 bit quantized and pruned models in Table 4 in our paper. Now, we also provide the inference time in seconds for all scale factors and bitwidths in the supplementary materials Table 11 and below.
> > ### Scale $\times 2$
> >
> > | Method  | Bit | Pruning | $s$    | MB    | $b$ |
> > |---------|-----|---------|--------|-------|-----|
> > | 2DQuant | 4   | No      | 0.040  | 13.92 | 4.0 |
> > | Ours    | 4   | No      | 0.084  | 13.92 | 4.0 |
> > | Ours    | 4   | Yes     | 0.084  | 13.87 | 3.4 |
> > | 2DQuant | 3   | No      | 0.040  | 13.84 | 3.0 |
> > | Ours    | 3   | No      | 0.084  | 13.84 | 3.0 |
> > | Ours    | 3   | Yes     | 0.084  | 13.82 | 2.8 |
> > | 2DQuant | 2   | No      | 0.039  | 13.75 | 2.0 |
> > | Ours    | 2   | No      | 0.086  | 13.76 | 2.0 |
> >
> > ---
> >
> > ### Scale $\times 3$
> >
> > | Method  | Bit | Pruning | $s$    | MB    | $b$ |
> > |---------|-----|---------|--------|-------|-----|
> > | 2DQuant | 4   | No      | 0.042  | 13.95 | 4.0 |
> > | Ours    | 4   | No      | 0.084  | 13.95 | 4.0 |
> > | Ours    | 4   | Yes     | 0.084  | 13.90 | 3.4 |
> > | 2DQuant | 3   | No      | 0.042  | 13.87 | 3.0 |
> > | Ours    | 3   | No      | 0.084  | 13.87 | 3.0 |
> > | Ours    | 3   | Yes     | 0.084  | 13.85 | 2.8 |
> > | 2DQuant | 2   | No      | 0.042  | 13.79 | 2.0 |
> > | Ours    | 2   | No      | 0.084  | 13.79 | 2.0 |
> >
> > ---
> >
> > ### Scale $\times 4$
> >
> > | Method  | Bit | Pruning | $s$     | MB    | $b$ |
> > |---------|-----|---------|---------|-------|-----|
> > | 2DQuant | 3   | No      | 0.0410  | 13.91 | 3.0 |
> > | Ours    | 3   | No      | 0.0883  | 13.91 | 3.0 |
> > | Ours    | 3   | Yes     | 0.0883  | 13.90 | 2.8 |
> > | 2DQuant | 2   | No      | 0.0410  | 13.83 | 2.0 |
> > | Ours    | 2   | No      | 0.0900  | 13.83 | 2.0 |
> >
> > signed‑rank tests for superiority: The signed‑rank tests for superiority vs. CondiQuant are conducted over the 5 most commonly chosen datasets for evaluating the performance of SR models.

---

> > > ### Author Response · Authors · 2025-12-03
> > >
> > > Regarding epsilon sensitivity, we clarify that the epsilon threshold is a way to observe and reason about the distribution of the weights and activations, not an actual part of the algorithm that will affect downstream PSNR. We wanted to observe if there was an increase in the proportion of values around 0 after the Hadamard Transform, which would allow us to prune those values or quantize them with lower errors. Therefore, we chose a small epsilon to observe this effect. To see whether we can still observe a statistically significant increase in the proportion of values in the [-epsilon,epsilon] region as we vary the values of epsilon, we repeated our analysis with epsilon from [0.01,1], testing whether the proportion of values increased for each 0.1 increment. The results are included in table 6 in the supplementary and also below. As we increase the value of epsilon, we see larger p-values and smaller effect sizes, which makes sense as many of these matrices are distributed around 0 with small standard deviations. However, many p-values are statistically significant, making this analysis not very sensitive to epsilon as long as it is sufficiently small.
> > >
> > > | $\epsilon$ | Weights $p$ | Weights $d_z$ | Acts $p$     | Acts $d_z$ |
> > > |------------|-------------|---------------|--------------|------------|
> > > | 0.01 | $1.1 \times 10^{-25}$ | 8.54 | $2.0 \times 10^{-8}$ | 0.72 |
> > > | 0.1  | $5.0 \times 10^{-22}$ | 1.38 | 0.0006 | 0.57 |
> > > | 0.2  | $2.5 \times 10^{-21}$ | 1.00 | 0.22   | 0.25 |
> > > | 0.3  | $1.2 \times 10^{-21}$ | 0.98 | 0.15   | 0.33 |
> > > | 0.4  | $2.8 \times 10^{-20}$ | 0.80 | 0.08   | 0.33 |
> > > | 0.5  | $5.6 \times 10^{-13}$ | 0.43 | 0.01   | 0.29 |
> > > | 0.6  | $2.5 \times 10^{-7}$  | 0.26 | 0.01   | 0.24 |
> > > | 0.7  | $1.7 \times 10^{-6}$  | 0.23 | 0.004  | 0.20 |
> > > | 0.8  | $9.9 \times 10^{-5}$  | 0.20 | 0.004  | 0.18 |
> > > | 0.9  | 0.001                 | 0.18 | 0.004  | 0.16 |
> > > | 1.0  | 0.01                  | 0.17 | 0.01   | 0.15 |
> > >
> > > Granularity: We have chosen per tensor level granularity, and we believe that increasing granularity would add more quantization scalars and zero points which will improve the performance, but in our method, we only add two parameters per tensor, not many more which would be what would happen with increased granularity. For example with per channel granularity, we would add more than 2, as any weight or activation matrix will have more than 2 channels.Therefore, finer granularity will help on its own, but it will add more parameters than we are adding now.
> > >
> > > Visual examples of methods with equal finetuning budget: All of our examples are against 2DQuant, the previous SOTA before CondiQuant, with equal finetuning budget. We also provide the details of the experiments in our github repository for exact reproducibility.
> > >
> > > Thank you so much for your consideration.

---

### Official Review · Reviewer_cUoK · 2025-11-05

**Soundness:** 1
**Presentation:** 1
**Contribution:** 1
**Rating:** 2
**Confidence:** 5

**Summary:**

This paper explores the use of Hadamard transformations for image super-resolution models, inspired by their demonstrated success in large language model (LLM) quantization.

**Strengths:**

* The paper targets an interesting direction.

**Weaknesses:**

The paper lacks a sufficiently comprehensive literature review and contains several inaccurate claims:
* At Line 73, the authors state that FlatQuant is a Hadamard-based method. This is incorrect — FlatQuant uses a learnable matrix instead of a fixed Hadamard transformation.
* The success of Hadamard or rotation-based transformations has already been extensively discussed in prior works [a, b, c]. These studies attribute the benefit to incoherent processing, as analyzed in QuIP[a]. However, the authors incorrectly claim that “the mechanism for Hadamard’s lowering of errors remains unexplored.”
* The concepts of learnable scale and offset have been explored in earlier works [d, e], so they cannot be considered novel contributions.
* The pruning component is overly simplistic, suggesting pruning parameters with values close to zero without discussing potential performance impacts or providing justification for the design choice. This section lacks insight and depth.

[a] Quip: 2-bit quantization of large language models with guarantees. NeurIPS 2023

[b] Quarot: Outlier-free 4-bit inference in rotated llms. NeurIPS 2024

[c] Spinquant: Llm quantization with learned rotations. ICLR 2025

[d] Learned step size quantization. ICLR 2020

[e] Lsq+: Improving low-bit quantization through learnable offsets and better initialization. CVPR 2020 workshop

**Questions:**

I strongly encourage the authors to conduct a more comprehensive and accurate literature review to better position this work within existing research.

---

> ### Author Response · Authors · 2025-11-21
>
> We thank you for taking the time to read our paper and provide feedback. We address your concerns below.
>
> Issue on line 73: On line 73, we stated that “Previous literature on quantization of LLMs Sun et al. (2024) introduced how “flatness” of weights and activations is important for quantization and Tseng et al. (2024) and Sun et al. (2024) stated that Hadamard transformations can increase flatness of distributions.” Our intention was not to imply that FlatQuant uses a Hadamard-based method, but rather that Sun et al. identify the importance of outlier removal (which they call “flatness”), and both Tseng et al. and Sun et al. cite prior work showing that Hadamard transforms reduce outliers as Sun et al. write: “Recent works find Hadamard … are particularly helpful in smoothing out outliers in activations (Xi et al., 2023; Ashkboos et al., 2024; Lin et al., 2024).” We have rewritten the passage to read: “Previous literature on quantization of LLMs (Ashkboos et al. (2024), Liu et al. (2024b), Federici et al. (2025), Chee et al. (2023), Sun et al. (2024), Tseng et al. (2024)) has found that outliers in weights and activations cause performance degradations. Sun et al. (2024) dubbed the removal of outliers as increasing the “flatness” of weights and activations and Federici et al. (2025), and Tseng et al. (2024) stated that Hadamard transformations can increase flatness of distributions.” Success of the Hadamard: Existing explanations of the Hadamard transform draw on different principles: quip and quip# emphasize incoherence processing; HadaNorm specifies the central limit theorem; and SpinQuant argues that random rotations blend large and small weights into smoother distributions. However, in quantization, there has not been an statistical analysis using activation and weight statistics to examine how the transform functions. To clarify this gap, we revised: “Therefore, the mechanism for the Hadamard’s lowering of the errors remains unexplored.” to: “Previous arguments on how the transform functions hinge on several concepts: incoherence processing, the central limit theorem and channel mixing. Tseng et al. (2024) found theorems that Hadamard transformations concentrate the entry magnitudes of distributions in a process called incoherence processing, and Liu et al. (2024b) stated that random rotations blend large and small weights together into a well-behaved distribution. Federici et al. (2025) cited the central limit theorem as to why the distributions after the Hadamard transform tend towards gaussian and thus have outliers reduced. However, a detailed empirical study of both activations and weights, including statistical significance tests of how these transforms affect outliers still remains underexplored."
>
> Difference from LSQ: Our method differs from LSQ/LSQ+ because we do not learn the quantization scale or zero point directly. Instead, the scale is determined by learned upper and lower bounds and the integer zero point corresponds to the learned lower bound. We additionally learn two novel parameters that modify the scale and zero point. To isolate their effect, we add an ablation where we freeze the upper/lower bounds and learn only our parameters on the 2-bit ×2 model (Table 7). From Figure 6 we know that the Hadamard transform alone does not surpass CondiQuant. However, in table 7 we see that only learning our parameters and not learning the upper and lower bounds improves performance, surpassing 2Dquant, but not surpassing CondiQuant. Thus, while beneficial, these parameters must be combined with learning the upper and lower bounds of the clipping range to have the combined effect of achieving the SOTA performance.
>
> Pruning section: Magnitude based pruning has been an established practice in compression literature. To emphasize this, the model pruning section in the related works now includes: “Regarding pruning criteria, we follow previous work Han et al. (2015), Lee et al. (2020), Li et al. (2018), Guo et al. (2016) in using Magnitude-based pruning, i.e. pruning the weight values with the smallest absolute magnitudes, assuming that they do not offer much information.” Furthermore, our pruning section in the methods now reads: “To achieve more compression at 3 and 4 bits, ..., we perform Magnitude-based pruning on the weights. Magnitude-based pruning suggests that weights with small magnitudes do not offer much information, and as shown in Figure 2, the transform allows for more values to be closer to 0, allowing us to remove them as they will have smallest absolute magnitudes to compress the weights further. …”

---

> ### Author Response · Authors · 2025-11-24
>
> Hello,
>
> We believe our rebuttal fixed concerns, and due to the coming deadline for the end of the discussion, we kindly remind the reviewer of the due date. We are happy to address further questions. Please let us know.
>
> Thank you

---

> > ### Comment · Reviewer_cUoK · 2025-11-26
> >
> > Thank you for the response.
> >
> > I would like to maintain my current score for the following reasons.
> >
> > First, I do not think there is any remaining gap to explore regarding the effectiveness of Hadamard-based methods. The idea that a reduced value range leads to smaller quantization error is well-established in the quantization literature. Activation quantization is known to be more challenging than weight quantization due to its wider dynamic range, which has motivated methods such as SmoothQuant and AWQ to explicitly compress activation ranges. Likewise, works such as QuaRot and SpinQuant have empirically demonstrated that rotation-based techniques reduce value ranges and therefore improve quantization errors. Given this strong foundation, I do not see meaningful unexplored space unless new, unexpected findings are presented.
> >
> > Second, the proposed scalar decomposition is essentially a modified version of LSQ. Without additional constraints, the learned $\alpha$ and $\beta$ could converge to equivalent behavior as LSQ+, making the approach theoretically similar and differing mainly in implementation. While the proposed initialization may be helpful, the paper provides no justification for why this design is fundamentally better beyond empirical outcomes, making it feel more like an engineering tweak rather than a substantive contribution.
> >
> > Third, the pruning component appears trivial. Without specialized hardware support, pruning parameters to zero does not yield practical speedups, and the improvement in storage is negligible, as shown in Table 4. Moreover, such pruning may harm model performance. As currently presented, the design is not well motivated.
> >
> > Overall, this work reads more like a technical report documenting the application of Hadamard-based quantization techniques to image super-resolution models. However, it lacks generalizable insights, conceptual novelty, or surprising findings that would justify publication as a formal academic contribution at ICLR.

---

> ### Author Response · Authors · 2025-11-26
>
> Regarding generalizability, we also benchmark our method on MambaIRv2-light, and achieve state of the art performance. These results are given in the supplementary section 7.7.
> Furthermore, Quarot does not provide empirical analysis and only cites Incoherence processing. Furthermore, SpinQuant only evaluates the kurtosis of activations, not any other property of activations and weights, and only plots the activations and weights, but does not provide any statistical analysis as to whether the difference in the kurtosis or plots is statistically significant. We provide statistical significance tests (wilcoxon signed rank tests) to show the effect of the Hadamard for both activations and weights, thus we feel that our work adds more insights into Hadamard transformations behavior.
>
> Regarding the pruning, we wanted to show that such a combination is possible without large performance degradations, and it opens up the possibility of further improvements. We now also benchmark our method on structured pruning of whole channels with the lowest absolute average magnitudes to reach the desired pruning percentage and show the results in Table 8 in the supplementary. We see that while this results in hardware compatibility, it has much worse performance than unstructured pruning.
>
> We also observe that the learned parameters do not converge to the same behavior as the learning the quantization scalar and zero points, since if that was the case then their performance would not surpass 2DQuant, but in Table 7 we see that our performance surpasses that of 2DQuant, but must be combined with learning the upper and lower bounds of the clipping range, which is not the same thing as learning the scalar but rather the same thing as learning the building blocks of the scalar to surpass CondiQuant and achieve SOTA, showing that both our parameters and the upper and lower bounds uniquely give their own benefits.

---

> ### Author Response · Authors · 2025-12-03
>
> Dear Area Chairs,
>
> Thank you so much for taking the time to read this and considering our paper. We would like to provide a brief, high-level summary of our contributions and clarify how we have addressed this reviewer's concerns during the rebuttal period.
> This reviewer raised an issue with the wording of our related works section, which we fully addressed and also raised the issue with novelty of our approach, citing that the Hadamard transformation's behavior has been fully studied. We push back against this by citing that our work is the only work to study with tests of statistical significance such as the wilcoxon signed rank test what changes the Hadamard transform induces and whether these changes in activations and weights are statistically significant and with what effect size, while prior work have only posited theories or empirical analysis only on activations with no tests of significance or plots of activations and weights pre and post transformation, again without any statistical significance analysis. We believe that we have pinned down the exact mechanisms for the Hadamard's effectiveness and show that these properties statistically significantly happen for both activations and weights, which is a more in depth and novel contribution in this space.
>
> Furthermore, this reviewer raised an issue regarding the novelty of our two additional parameters which modify the quantization scalar and zero points. We push back against this by first stating that we do not learn the quantization scalar and zero point directly like in LSQ/LSQ+, and instead we only learn the upper and lower bounds of the clipping range (which was the main contribution of 2DQuant), in addition to our parameters. We also show with the ablation study on the 2-bit quantized SwinIr-light $\\times 2$ model below that the two additional parameters we added, which modify the scalar after it has been calculated from the upper and lower bounds and the zero point which is instantiated as the lower bound, do not in fact converge to learning the scalar and zero points directly.
> | Method ($\times 4$)                  | Bit | PSNR  | SSIM   |
> |--------------------------------------|-----|-------|--------|
> | 2DQuant                              | 2   | 36.00 | 0.9497 |
> | CondiQuant                           | 2   | 37.15 | 0.9567 |
> | Ours – $\alpha,\beta$ tuning only    | 2   | 36.50 | 0.9519 |
> | Ours – tuning all                    | 2   | 38.03 | 0.9605 |
>
> Here we see that only learning our parameters surpasses the performance of 2DQuant, but if our parameters converged to learning the scalar then their performance would not surpass 2DQuant. We also see that our approach must be combined with learning the upper and lower bounds of the clipping range to surpass CondiQuant and achieve SOTA, showing that both our parameters and the upper and lower bounds uniquely give their own benefits, and do not converge to the same behavior. This table is provided in the supplementary materials as well.
>
> Regarding the pruning, the reviewer cited doubts regarding our motivation and design, but we show that Magnitude based pruning has been a standard practice in compression literature, and regarding hardware compatibility, we now also benchmark our method on structured pruning of whole channels with the lowest absolute average magnitudes to reach the desired pruning percentage and show the results below.
> | Method ($\times 4$) | Bit | $b$ | MB    | $s$   | PSNR  | SSIM   |
> |---------------------|-----|----:|------:|------:|-------|--------|
> | Ours                | 4   | 3.4 | 13.95 | 0.092 | 32.02 | 0.8900 |
> | Ours                | 4   | 2.4 | 13.86 | 0.120 | 29.92 | 0.8471 |
> | Ours                | 3   | 2.8 | 13.90 | 0.088 | 31.74 | 0.8900 |
> | Ours                | 3   | 1.8 | 13.82 | 0.130 | 29.90 | 0.8462 |
> | Ours                | 2   | 2.0 | 13.83 | 0.090 | 31.44 | 0.8820 |
> | Ours                | 2   | 1.2 | 13.77 | 0.120 | 29.84 | 0.8460 |
>
> Here we see that the structured pruning approach results in more performance degradation and slower inference time, given by $s$, which is why we did not focus on it in our work, and instead decided to combine our method with the more performant pruning method, to showcase feasibility of combining quantization and pruning on SwinIr-light due to the effects of the Hadamard transformation. This table is also provided in the supplementary materials.
>
> We truly appreciate your time and consideration.

---

### Meta-Review · Area_Chair_QchN · 2026-01-05

**Summary:**

The submission proposes CompSRT, a compression pipeline for Image Super-Resolution (SR) Transformers. The method combines Hadamard-based quantization with a scalar decomposition technique that introduces two additional trainable parameters to refine the quantization scale and zero-point.

Reviewers identified that some core components like learnable quantization parameters are well-established in quantization literature. The reviewers found the application of these known techniques to the SR domain to be weak scientific novelty.

Given the above unaddressed concerns, this submission cannot be accepted in the current form.

**Reviewer Concerns:**

The rebuttal successfully addressed several empirical and clarifying concerns, particularly regarding the fairness of comparisons and the generalizability of the results. The authors provided new experiments on MambaIRv2-light to prove the method’s broad applicability and clarified that their performance gains were not merely a result of a superior training recipe, as their fine-tuning budget was identical to the 2DQuant baseline. They also offered a more rigorous statistical defense of the Hadamard transform’s behavior, moving beyond the intuitive "incoherence" arguments of prior work.

However, significant concerns regarding conceptual novelty and practical utility remain outstanding. Reviewers noted that some core components like Hadamard-based outlier reduction and learnable scaling parameters closely resemble existing techniques, leading to the assessment that the work is more of an incremental engineering application than a fundamental contribution. Furthermore, the unstructured pruning component was deemed trivial and lacking in hardware-specific justification, as it offers negligible storage benefits and no practical speedup without specialized kernels.

**Reviewer Scores:**

The reviewers’ scores likely would have polarized further had the discussion continued to its full conclusion. Reviewer cUoK and VCpf  remained firmly at the initial scores,  as the rebuttal failed to shift their perception that the work is an incremental application of existing principles rather than a novel scientific advancement.

---

### Decision · Program_Chairs · 2026-01-26

Reject